# Do you know what k-means? Clustering with constant number of samples

## Abstract

Clustering is one of the most important tools for analysis of large datasets, and perhaps the most popular clustering algorithm is Lloyd's algorithm for $k$-means. This algorithm takes $n$ vectors $V = [v_1, \ldots, v_n] \in \mathbb{R}^{d \times n}$ and outputs $k$ centroids $c_1, \ldots, c_k \in \mathbb{R}^d$; these partition the vectors into clusters based on which centroid is closest to a particular vector. We present a classical $\varepsilon$-$k$-means algorithm that performs an approximate version of one iteration of Lloyd's algorithm with time complexity $\widetilde{O}\big(\frac{\|V\|_F^2}{n} \frac{k^2 d}{\varepsilon^2}(k + \log n)\big)$, exponentially improving the dependence on the data size $n$ and matching that of the "$q$-means" quantum algorithm originally proposed by Kerenidis, Landman, Luongo, and Prakash (NeurIPS'19). Moreover, we propose an improved $q$-means quantum algorithm with time complexity $\widetilde{O}\big(\frac{\|V\|_F}{\sqrt{n}} \frac{k^{3/2} d}{\varepsilon}(\sqrt{k} + \sqrt{d})(\sqrt{k} + \log n)\big)$ that quadratically improves the runtime of our classical $\varepsilon$-$k$-means algorithm in several parameters. Our quantum algorithm does not rely on quantum linear algebra primitives of prior work, but instead only uses QRAM to prepare simple states based on the current iteration's clusters and multivariate quantum mean estimation. Our upper bounds are complemented with classical and quantum query lower bounds, showing that our algorithms are optimal in most parameters. Finally, we conduct numerical experiments that evidence the substantially improved runtime our classical algorithm over the standard Lloyd's algorithm, thus being one of the first cases of a practical dequantised algorithm.

## 1 Introduction

Among machine learning problems, data clustering and the $k$-means problem are of particular relevance and have attracted much attention in the past (Hartigan & Wong, 1979; Krishna & Murty, 1999; Likas et al., 2003). Here the task is to find an assignment of each vector from a dataset of size $n$ to one of $k$ labels (for a given $k$ assumed to be known) such that similar vectors are assigned to the same cluster. To be more precise, in the $k$-means problem we are given $n$ vectors $v_1, \ldots, v_n \in \mathbb{R}^d$ as columns in a matrix $V \in \mathbb{R}^{d \times n}$, and a positive integer $k$, and the task is to find $k$ centroids $c_1, \ldots, c_k \in \mathbb{R}^d$ such that the cost function $\sum_{i \in [n]} \min_{j \in [k]} \|v_i - c_j\|^2$, called *residual sum of squares*, is minimized, where $\|v_i - c_j\|$ is the Euclidean distance between $v_i$ and $c_j$ and $[n] := \{1, \ldots, n\}$ for $n \in \mathbb{N} := \{1, 2, \ldots\}$.

Since the $k$-means problem is known to be NP-hard (Dasgupta, 2008; Vattani, 2009; Mahajan et al., 2012), several classical polynomial-time algorithms have been developed to obtain *approximate* solutions (Kanungo et al., 2002; Jaiswal et al., 2014; Ahmadian et al., 2017; Bhattacharya et al., 2020). One such algorithm is the $k$-means algorithm (also known as Lloyd's algorithm) introduced by Lloyd (1982), a heuristic algorithm that iteratively updates the centroids $c_1, \ldots, c_k$ until some desired precision is reached. At each time step $t$, the algorithm clusters the data into $k$ clusters denoted by the sets $\mathcal{C}_j^t \subseteq [n]$, $j \in [k]$, each with centroid $c_j^t$, and then updates the centroids based on such clustering. More precisely, the $k$-means algorithm starts with initial centroids $c_1^0, \ldots, c_k^0 \in \mathbb{R}^d$, picked either randomly or through some pre-processing routine as in the $k$-means++ algorithm (Arthur & Vassilvitskii, 2007), and alternates between two steps:

1. Each vector $v_i$, $i \in [n]$, is assigned a cluster $\mathcal{C}_{\ell_i^t}^t$ where $\ell_i^t = \arg\min_{j \in [k]} \|v_i - c_j^t\|$;

2. The new centroids $\{c_j^{t+1}\}_{j\in[k]}$ are updated based on the new clusters, $c_j^{t+1} = \frac{1}{|\mathcal{C}_j^t|}\sum_{i\in\mathcal{C}_j^t} v_i$.

The above steps are repeated until $\frac{1}{k}\sum_{j\in[k]} \|c_j^t - c_j^{t+1}\| \leq \tau$ for a given threshold $\tau > 0$, in which case we say that the algorithm has converged. The naive runtime of a single iteration is $O(nkd)$.

On the other hand, the subfield of quantum machine learning aims to offer computational advantages in machine learning, with many new proposed algorithms (Lloyd et al., 2014; Kerenidis & Prakash, 2017; 2020a; Chakraborty et al., 2019; Lloyd et al., 2013; Allcock et al., 2020; Ambainis et al., 2025). A notable line of work in this subfield is quantum versions of the $k$-means algorithm (Aïmeur et al., 2013; Lloyd et al., 2013). More notably, Kerenidis et al. (2019) proposed a quantum version of Lloyd's algorithm called $q$-means. They use quantum linear algebra subroutines and assume the input vectors are stored in QRAM (Quantum Random Access Memory) (Giovannetti et al., 2008a;b) and that all clusters are of roughly equal size. Their per-iteration runtime depends only polylogarithmically on the size $n$ of the dataset, an exponential improvement compared to prior works. However, their quantum algorithm only performs each iteration approximately. The authors showed that $q$-means performs a robust version of $k$-means called $(\varepsilon, \nu)$-$k$-means, and then showed through experiments that the approximation does not worsen the quality of the centroids. In the $(\varepsilon, \nu)$-$k$-means algorithm, two noise parameters $\varepsilon, \nu \geq 0$ are introduced in the distance estimation and centroid update steps. It alternates between the steps:

1. A vector $v_i$ is assigned a cluster $\mathcal{C}_{\ell_i^t}^t$ where $\ell_i^t \in \{j \in [k] : \|v_i - c_j^t\|^2 \leq \min_{j'\in[k]} \|v_i - c_{j'}^t\|^2 + \nu\}$;

2. The new centroids $\{c_j^{t+1}\}_{j\in[k]}$ are updated such that $\left\| c_j^{t+1} - \frac{1}{|\mathcal{C}_j^t|}\sum_{i\in\mathcal{C}_j^t} v_i \right\| \leq \varepsilon$.

The overall idea behind $q$-means from Kerenidis et al. (2019) is to first create the quantum state $n^{-1/2}\sum_{i\in[n]} |i\rangle|\ell_i^t\rangle$ using distance estimation and quantum minimum finding (Dürr & Høyer, 1996), followed by measuring the label register $|\ell_i^t\rangle$ to obtain $|\mathcal{C}_j^t|^{-1/2}\sum_{i\in\mathcal{C}_j^t} |i\rangle$ for some random $j \in [k]$. The $q$-means algorithm then proceeds to perform a matrix multiplication with matrix $V$ by using quantum linear algebra techniques, followed by quantum tomography (Kerenidis & Prakash, 2020b) in order to retrieve a classical description. Since quantum linear algebra techniques are used, the final runtime depends on quantities like the condition number $\kappa(V)$ of the matrix $V$ and a matrix dependent parameter $\mu(V)$ which is upper-bounded by the Frobenius norm $\|V\|_F$ of $V$.

**Fact 1** ((Kerenidis et al., 2019, Theorem 3.1)). *For $\varepsilon > 0$ and dataset matrix $V \in \mathbb{R}^{d\times n}$ with $\|V\|_{2,\infty} := \max_{i\in[n]} \|v_i\| \geq 1$ and condition number $\kappa(V)$, the $q$-means algorithm outputs centroids consistent with the $(\varepsilon, \nu)$-$k$-means algorithm in $\widetilde{O}\big(\frac{kd}{\varepsilon^2}\|V\|_{2,\infty}^2\kappa(V)(\mu(V) + \frac{k}{\nu}\|V\|_{2,\infty}^2) + \frac{k^2}{\varepsilon\nu}\|V\|_{2,\infty}^3\kappa(V)\mu(V)\big)$ time per iteration.*

In this work, we provide *exponentially* improved classical and quantum $(\varepsilon, \nu)$-$k$-means algorithms that match the logarithmic dependence on $n$ of the $q$-means algorithm from Kerenidis et al. (2019), while substantially improving the dependence on other parameters.

## 2 COMPUTATIONAL MODELS

For $x \in \mathbb{R}^d$, let $\|x\|_r = (\sum_{i\in[d]} |x_i|^r)^{\frac{1}{r}}$ for $r \in [1, \infty]$ and $\|x\| = \|x\|_2$. Let $\mathcal{D}_x^{(1)}$ and $\mathcal{D}_x^{(2)}$ be the distributions over $[d]$ with probability density functions $\mathcal{D}_x^{(1)}(i) = \frac{|x_i|}{\|x\|_1}$ and $\mathcal{D}_x^{(2)}(i) = \frac{x_i^2}{\|x\|^2}$. For $V \in \mathbb{R}^{d\times n}$, let the matrix norms:

- $\|V\| = \max_{x\in\mathbb{R}^n:\|x\|=1} \|Vx\|$ (spectral norm);
- $\|V\|_F = (\sum_{i\in[n]}\sum_{l\in[d]} V_{li}^2)^{\frac{1}{2}}$ (Frobenius norm);
- $\|V\|_{1,1} = \sum_{i\in[n]}\sum_{l\in[d]} |V_{li}| = \sum_{i\in[n]} \|v_i\|_1$;
- $\|V\|_{2,1} = \sum_{i\in[n]} \|v_i\|$;
- $\|V\|_{2,\infty} = \max_{i\in[n]} \|v_i\|$.

It is known that $\|V\|_{2,\infty} \leq \|V\| \leq \|V\|_F \leq \|V\|_{2,1} \leq \|V\|_{1,1}$ and $\|V\|_F \leq \sqrt{\min\{n,d\}}\|V\| \leq \sqrt{n\min\{n,d\}}\|V\|_{2,\infty}$ and $\|V\|_{1,1} \leq \sqrt{d}\|V\|_{2,1} \leq \sqrt{nd}\|V\|_F \leq n\sqrt{d}\|V\|_{2,\infty}$. Let $\mathcal{D}_V^{(1)}$ be the distribution over $[n]\times[d]$ with probability density function $\mathcal{D}_V^{(1)}(i, l) = \frac{|V_{li}|}{\|V\|_{1,1}}$.

Little background on quantum computing is required for our paper and we point the reader to Nielsen & Chuang (2010) for more information. A quantum system is described by a unit vector from a complex Hilbert space, denoted by the ket notation $|\cdot\rangle$. A qubit, the quantum equivalent of a bit, is a quantum system described by a unit vector in $\mathbb{C}^2$, while an $n$-qubit system is described by a unit vector in $\mathbb{C}^{2^n}$. The evolution of a quantum state $|\psi\rangle \in \mathbb{C}^{2^n}$ is described by a unitary operator, or quantum gate, $U \in \mathbb{C}^{2^n \times 2^n}$ such that $UU^\dagger = I$ where $U^\dagger$ is the Hermitian conjugate of $U$. In order to extract classical information from a quantum system, a quantum measurement is performed, which is a set $\{E_m\}_m$ of positive operators $E_m \succ 0$ that sum to identity, $\sum_m E_m = I$. The probability of measuring $E_m$ on $|\psi\rangle$ is $\langle\psi|E_m|\psi\rangle$. We let $|\bar{0}\rangle$ denote the state $|0\rangle \otimes \cdots \otimes |0\rangle$ where the number of qubits is clear from context.

In this work, we assume a standard computational model — wherein arithmetic operations require $O(1)$ time — enhanced by a special low-overhead classical/quantum data structures that allows for efficiently querying and sampling the dataset and centroid matrices $V \in \mathbb{R}^{d \times n}$ and $C \in \mathbb{R}^{d \times k}$.

**Definition 2** (Classical query access). *We say we have* (classical) query access *to a matrix* $V = [v_1, \ldots, v_n] \in \mathbb{R}^{d \times n}$ *if $V$ is stored in a data structure that supports the following operations:*

    *1. Reading an entry of $V$ in $O(\log(nd))$ time;*

    *2. Finding $\|v_i\|$ in $O(\log n)$ time for any given $i \in [n]$;*

    *3. Finding $\|V\|_{2,1}$ or $\|V\|_{1,1}$ in $O(1)$ time;*

    *4. Sampling from $\mathcal{D}^{(1)}_{(\|v_j\|)_{j=1}^n}(i) = \frac{\|v_i\|}{\|V\|_{2,1}}$ in $O(\log n)$ time;*

    *5. Sampling from $\mathcal{D}^{(2)}_{v_i}(l) = \frac{V_{li}^2}{\|v_i\|^2}$ or $\mathcal{D}^{(1)}_V(i,l) = \frac{|V_{li}|}{\|V\|_{1,1}}$ in $O(\log(nd))$ time.*

**Definition 3** (Quantum query access). *We say we have* quantum query access *to a matrix* $V = [v_1, \ldots, v_n] \in \mathbb{R}^{d \times n}$ *if $V$ is stored in a data structure that supports the following operations:*

    *1. Reading an entry of $V$ in $O(\log(nd))$ time;*

    *2. Finding $\|v_i\|$ in $O(\log n)$ time for any given $i \in [n]$;*

    *3. Finding $\|V\|_{2,1}$ or $\|V\|_{1,1}$ in $O(1)$ time;*

    *4. Mapping $|\bar{0}\rangle \mapsto \sum_{i \in [n]} \sqrt{\frac{\|v_i\|}{\|V\|_{2,1}}}|i\rangle$ in $O(\log n)$ time;*

    *5. Mapping $|\bar{0}\rangle \mapsto \sum_{(i,l) \in [n] \times [d]} \sqrt{\frac{|V_{li}|}{\|V\|_{1,1}}}|i,l\rangle$ in $O(\log(nd))$ time;*

    *6. Mapping $|i\rangle|\bar{0}\rangle \mapsto \sum_{l \in [d]} \frac{V_{li}}{\|v_i\|}|i,l\rangle$ in $O(\log(nd))$ time;*

    *7. Mapping $|i,l\rangle|\bar{0}\rangle \mapsto |i,l\rangle|V_{li}\rangle$ in $O(\log(nd))$ time;*

    *8. Mapping $|i\rangle|\bar{0}\rangle \mapsto |i\rangle|v_i\rangle = |i\rangle|V_{1i}, \ldots, V_{di}\rangle$ in $O(d\log(nd))$ time.*

In our classical computation model, all arithmetic operations require $O(1)$ time. We refer to any operation in Definition 2 as a *classical query*. Our computational model thus assumes classical query access to $V$, meaning that the dataset matrix has been pre-processed beforehand and all operations from Definition 2 can be performed with their respective stated time complexities. The pre-processing phase, which takes $\widetilde{O}(nd)$ time, basically requires computing all the norms $\{\|v_i\|\}_{i \in [n]}$, $\|V\|_{2,1}$, and $\|V\|_{1,1}$, plus inserting $V$ into a RAM structure in order to efficiently read any of its entries and into specialised binary trees (Prakash, 2014; Kerenidis & Prakash, 2017) in order to efficiently sample from the distributions $\mathcal{D}^{(1)}_{(\|v_i\|)_{i=1}^n}$, $\mathcal{D}^{(2)}_{v_i}$, and $\mathcal{D}^{(1)}_V$. The centroid matrix $C \in \mathbb{R}^{d \times k}$ is *not* assumed to be pre-processed, since it changes throughout the $k$-means algorithm, and thus, if we ever require classical query access to $C$, we must pay the pre-processing price of $\widetilde{O}(kd)$ time. We note that norm sampling is a well-established technique in machine learning (Hazan et al., 2011; Song et al., 2016) and randomised linear algebra (Frieze et al., 2004; Drineas et al., 2008; Kannan & Vempala, 2017), so the data structures from Definition 2 are reasonable. Finally, we define a measure of (classical) *time complexity* as the sum of the times of all arithmetic and classical queries comprising some given computation. In other words, if a computation is composed of $m$ arithmetic operations and $q$ classical queries, its time complexity is at most $O(m + q\log(nd))$.

In our quantum computation model, all single and two-qubit quantum gates require $O(1)$ time, and more generally, all arithmetic operations require $O(1)$ time. We refer to any operation in Definition 3 as a *quantum query*, except Item 8 which comprises $d$ quantum queries. Our quantum model assumes quantum query access to the dataset matrix $V$, meaning that it has been pre-processed beforehand and all operations from Definition 3 can be performed with their respective stated time complexities. In Definition 3, the unitaries from Items 7 and 8 are known as a quantum random access memory (QRAM) (Giovannetti et al., 2008a;b) and can be seen as the quantum equivalent of a classical RAM. As part of the pre-processing stage, we assume that $V$ has been ported into a QRAM structure, for which several several proposals exist (Giovannetti et al., 2008a;b; Zoufal et al., 2019; Park et al., 2019; Hann et al., 2019; Chen et al., 2021; Niu et al., 2022; Phalak et al., 2022; Niu et al., 2022; Agliardi & Prati, 2022; Allcock et al., 2024; Wang et al., 2025); see Hann (2021); Phalak et al. (2023); Jaques & Rattew (2023) for a few surveys. In this work, we simply assume that a QRAM requires a running time proportional to its circuit depth, i.e., logarithmitically in the memory size it accesses. Although a somewhat controversial resource, there have been recent results which suggest that the cost of QRAM is much smaller than previously assumed (Hann et al., 2021; Mehta et al., 2024; Dalzell et al., 2025). Besides, an experimental QRAM implementation have been recently reported in Shen et al. (2025). Finally, QRAM mimics the classical RAM model in the quantum setting as closely as possible and we thus view it as a fair comparison to our classical model.

The classical data structure that allows the operations from Items 4 to 6 is usually known as a KP-tree and was proposed in Prakash (2014); Kerenidis & Prakash (2017). In a nutshell, a KP-tree is a rooted binary tree which contains the partial sums of the entries of a given vector in its nodes. We note that a KP-tree can also be used to performed the classical sampling operations from Definition 2. Finally, we define a measure of (quantum) *time complexity* as the sum of the times of all arithmetic and quantum queries comprising some given computation. In other words, if a computation is composed of $m$ arithmetic operations and $q$ quantum queries, its time complexity is at most $O(m + q \log(nd))$.

## 3 OUR ALGORITHMS

**Classical algorithms.** In this work, we provide *exponentially* improved classical $(\varepsilon, \nu)$-$k$-means algorithms that match the logarithmic dependence on $n$ of the $q$-means algorithm from Kerenidis et al. (2019). Our first classical algorithm, named EKMeans[1] and described in Algorithm 1, is an approximate version of the standard $k$-means algorithm wherein the quantities $\sum_{i \in \mathcal{C}_j^t} v_i$ and $|\mathcal{C}_j^t|$ for each $j \in [k]$ are estimated using the classical query access of Definition 2, from which the new centroids $c_j^{t+1} \approx |\mathcal{C}_j^t|^{-1} \sum_{i \in \mathcal{C}_j^t} v_i$ can be approximated. In more details, the cluster sizes $|\mathcal{C}_j^t|$ are estimated by first sampling a set of indices $P \subseteq [n]$ uniformly at random, while the sums $\sum_{i \in \mathcal{C}_j^t} v_i$ are estimated by first sampling a set of indices $Q \subseteq [n]$ from the distribution $\mathcal{D}_{(\|v_i\|)_{i=1}^n}^{(1)}$, i.e., by $\ell_1$-sampling from the vector of $V$'s column $\ell_2$-norms. EKMeans then mimics the standard $k$-means algorithm in that it finds the closest centroid to each vector $v_i$, $i \in P \cup Q$, by exactly computing $\|v_i - c_j^t\|^2$ in $O(d)$ time. We then prove that the subset $P_j \subset P$ of sampled vectors closest to $c_j^t$ well approximates $\frac{|P_j|}{|P|} \approx \frac{1}{n}|\mathcal{C}_j^t|$ with high probability. Likewise, the subset $Q_j \subset Q$ of sampled vectors closest to $c_j^t$ can be used to approximate $\sum_{i \in Q_j} \frac{\|V\|_{2,1}}{|Q|} \frac{v_i}{\|v_i\|} \approx \sum_{i \in \mathcal{C}_j^t} v_i$ with high probability. By assuming that all clusters $\{\mathcal{C}_j^t\}_{j \in [k]}$ are of roughly the same size, we can show that the number of required samples is independent of $n$. The precise query and time complexities of EKMeans are described in Table 1 below, while its proof is postponed to Appendix A. Interestingly enough, the proof of correctness employs the advanced Freedman's inequality for martingales (see Fact 6). Note that EKMeans is consistent with a $(\varepsilon, \nu=0)$-$k$-means algorithm since the cluster assignment is done exactly and that it requires the operations from Items 1 to 4 of Definition 2.

The dependence on $n$ comes from sampling the indices $P, Q$ in $O((|P| + |Q|) \log n)$ time under Definition 2, which is performed once before every iteration. In practice, it is possible to draw $P, Q$ only once before the first iteration and reuse the same samples throughout the clustering procedure by slightly increasing the sizes of $P, Q$. More precisely, a union bound over all $T$ iterations leads

---

[1]Pronounced [i:k mi:nz].

---

**Algorithm 1** Classical $(\varepsilon, \nu = 0)$-$k$-means algorithm EKMeans

---

**Input:** Classical query access to data matrix $V = [v_1, \dots, v_n] \in \mathbb{R}^{d \times n}$, parameters $\delta, \varepsilon$.

1: Select $k$ initial centroids $c_1^0, \dots, c_k^0$
2: **for** $t = 0$ until convergence **do**
3:      Sample $p = O\big(\frac{\|V\|^2}{n} \frac{k^2}{\varepsilon^2} \log \frac{k}{\delta}\big)$ indices $P \subseteq [n]$ uniformly from $[n]$
4:      Sample $q = O\big(\frac{\|V\|_{2,1}^2}{n^2} \frac{k^2}{\varepsilon^2} \log \frac{k}{\delta}\big)$ indices $Q \subseteq [n]$ from $\mathcal{D}_{(\|v_i\|)_{i=1}^n}^{(1)}$
5:      For $i \in P \cup Q$, label $\ell_i^t = \arg\min_{j \in [k]} \|v_i - c_j^t\|$
6:      For $j \in [k]$, let $P_j := \{i \in P | \ell_i^t = j\}$ and $Q_j := \{i \in Q | \ell_i^t = j\}$
7:      For $j \in [k]$, let the new centroids $c_j^{t+1} = \frac{p}{n|P_j|} \sum_{i \in Q_j} \frac{\|V\|_{2,1}}{q} \frac{v_i}{\|v_i\|}$
8: **end for**

---

to $P, Q$ which are good enough for all iterations with probability $1 - \delta$ by sampling a factor of $\log(T/\delta)/\log(1/\delta)$ more indices. The time dependence on $n$ is thus amortised over all iterations.

We then propose a second classical algorithm (Algorithm 2 in Appendix A) similar to Algorithm 1 but where the distances $\|v_i - c_j^t\|^2$ are now approximated up to error $\frac{\nu}{2}$ via an $\ell_2$-sampling procedure (Lemma 11), which yields the approximate labels $\ell_i^t \in \{j \in [k] : \|v_i - c_j^t\|^2 \le \min_{j' \in [k]} \|v_i - c_{j'}^t\|^2 + \nu\}$. The $\ell_2$-sampling subroutine to estimate $\|v_i - c_j^t\|^2$, which has been employed before in Tang (2019), requires sampling from the distribution $\mathcal{D}_{v_i}^{(2)}$ and uses a median-of-means estimator. Another main difference from Algorithm 1 is that our $(\varepsilon, \nu)$-$k$-means Algorithm 2 samples entries of $V$ via the distribution $\mathcal{D}_V^{(1)}$ instead of sampling columns of $V$ via the distribution $\mathcal{D}_{(\|v_i\|)_{i=1}^n}^{(1)}$ in order to improve the complexity dependence on $d$ (at the cost of worsening the dependence on other parameters). The precise query and time complexities of Algorithm 2 are described in Table 1 below, while its analysis is postponed to Appendix A. Note that Algorithm 2 requires the operations from Items 1 to 3 and 5 of Definition 2.

We note that Algorithms 1 and 2 can be viewed as a type of mini-batch $k$-means algorithm (Sculley, 2010; Bejarano et al., 2011), where instead of uniformly selecting a subset of the dataset, we leverage the query access from Definition 2 to sample a subset that depends on the norm distribution of $v_1, \dots, v_n$, yielding better sample and time complexities (see Theorem 9).

**Quantum algorithms.** Beyond classical algorithms, we also propose improved *quantum* algorithms that avoid the need for quantum linear algebra subroutines as in Kerenidis et al. (2019) and still keep the logarithmic dependence on the size $n$ of the dataset, while *improving* the complexity of the original $q$-means and of our classical $(\varepsilon, \nu)$-$k$-means algorithms in several parameters. Similar to our classical algorithms, we approximate separately the quantities $|\mathcal{C}_j^t|$ and $\sum_{i \in \mathcal{C}_j^t} v_i$ for $j \in [k]$, but now employing inherently quantum subroutines. Similar to Kerenidis et al. (2019), our first quantum algorithm (Algorithm 3 in Appendix B) constructs states of the form $\sum_{i \in [n]} \frac{1}{\sqrt{n}} |i\rangle$ and $\sum_{i \in [n]} \sqrt{\frac{\|v_i\|}{\|V\|_{2,1}}} |i\rangle$ using quantum query access to $V$ from Definition 3. By quantumly calling a classical circuit to exactly compute the distances $\|v_i - c_j^t\|^2$ in $O(d)$ time (how to do so is standard in quantum computing, see Nielsen & Chuang (2010)) plus the quantum minimum finding subroutine from Dürr & Høyer (1996) to find the minimum $\ell_i^t = \arg\min_{j \in [k]} \|v_i - c_j^t\|$ in $O(\sqrt{k}d)$ quantum queries, we then obtain the states $\sum_{i \in [n]} \frac{1}{\sqrt{n}} |i, \ell_i^t\rangle$ and $\sum_{i \in [n]} \sqrt{\frac{\|v_i\|}{\|V\|_{2,1}}} |i, \ell_i^t\rangle$. Up to this point, Algorithm 3 behaves similarly to $q$-means from Kerenidis et al. (2019), but instead of performing quantum linear algebra transformations and quantum tomography, we input such states (or more precisely the unitaries behind them) into the multivariate quantum mean estimator from Cornelissen et al. (2022). Their subroutine, although highly non-trivial, basically outputs an estimate to the mean $\sum_{\omega \in \Omega} \mathbb{P}(\omega) X(\omega)$ of some multivariate random variable $X : \Omega \to \mathbb{R}^N$ over a probability space $(\Omega, 2^\Omega, \mathbb{P})$ by assuming access to the unitaries $|\bar{0}\rangle \mapsto \sum_{\omega \in \Omega} \sqrt{\mathbb{P}(\omega)} |\omega\rangle$ and $|\omega, \bar{0}\rangle \mapsto |\omega, X(\omega)\rangle$. Applied to our case, the state $\sum_{i \in [n]} \frac{1}{\sqrt{n}} |i, \ell_i^t\rangle$ encodes the uniform probability distribution over $[n]$ used to approximate $|\mathcal{C}_j^t|$, while $\sum_{i \in [n]} \sqrt{\frac{\|v_i\|}{\|V\|_{2,1}}} |i, \ell_i^t\rangle$ encodes the probability distribution

Table 1: Query and time complexities per iteration of our classical and quantum algorithms assuming $|\mathcal{C}_j^t| = \Omega(\frac{n}{k})$ for all $j \in [k]$. The error $(\varepsilon, \nu)$ refer to the error $\nu$ in assigning a vector $v_i$ to a cluster $\mathcal{C}_{\ell_i^t}^t$ with $\ell_i^t \in \{j \in [k] : \|v_i - c_j^t\|^2 \le \min_{j' \in [k]} \|v_i - c_{j'}^t\|^2 + \nu\}$ and the error $\varepsilon$ in computing the new centroids as $\left\| c_j^{t+1} - |\mathcal{C}_j^t|^{-1} \sum_{i \in \mathcal{C}_j^t} v_i \right\| \le \varepsilon$. The matrix norms are $\|V\| = \max_{x \in \mathbb{R}^d : \|x\| = 1} \|Vx\|$, $\|V\|_F = (\sum_{i \in [n], l \in [d]} V_{li}^2)^{\frac{1}{2}}$, $\|V\|_{1,1} = \sum_{i \in [n], l \in [d]} |V_{li}|$, $\|V\|_{2,1} = \sum_{i \in [n]} \|v_i\|$, $\|V\|_{2,\infty} = \max_{i \in [n]} \|v_i\|$. The quantum runtimes can be slightly improved given access to a special gate called QRAG (Ambainis, 2007; Allcock et al., 2024) (see Footnotes 4 and 5). All complexities are up to polylog factors in $k$, $d$, $\frac{1}{\varepsilon}$, $\frac{1}{\nu}$, $\frac{\|V\|_F}{\sqrt{n}}$.

| Alg. | Error | Query complexity | Time complexity |
|---|---|---|---|
| Class. Alg. 1 | $(\varepsilon, 0)$ | $\left( \frac{\|V\|^2}{n} + \frac{\|V\|_{2,1}^2}{n^2} \right) \frac{k^2 d}{\varepsilon^2}$ | $\left( \frac{\|V\|^2}{n} + \frac{\|V\|_{2,1}^2}{n^2} \right) \frac{k^2 d}{\varepsilon^2} (\log n + k)$ |
| Class. Alg. 2 | $(\varepsilon, \nu)$ | $\left( \frac{\|V\|^2}{n} + \frac{\|V\|_{1,1}^2}{n^2} \right) \frac{\|V\|_F^2 \|V\|_{2,\infty}^2}{n} \frac{k^3}{\varepsilon^2 \nu^2}$ | $\left( \frac{\|V\|^2}{n} + \frac{\|V\|_{1,1}^2}{n^2} \right) \frac{\|V\|_F^2 \|V\|_{2,\infty}^2}{n} \frac{k^3}{\varepsilon^2 \nu^2} \log n$ |
| Quant. Alg. 3 | $(\varepsilon, 0)$ | $\left( \sqrt{k} \frac{\|V\|}{\sqrt{n}} + \sqrt{d} \frac{\|V\|_{2,1}}{n} \right) \frac{k^{\frac{3}{2}} d}{\varepsilon}$ | $\left( \sqrt{k} \frac{\|V\|}{\sqrt{n}} + \sqrt{d} \frac{\|V\|_{2,1}}{n} \right) \frac{k^{\frac{3}{2}} d}{\varepsilon} (\log n + \sqrt{k})$ |
| Quant. Alg. 4 | $(\varepsilon, \nu)$ | $\left( \sqrt{k} \frac{\|V\|}{\sqrt{n}} + \sqrt{d} \frac{\|V\|_{1,1}}{n} \right) \frac{\|V\|_F \|V\|_{2,\infty}}{\sqrt{n}} \frac{k^{\frac{3}{2}}}{\varepsilon \nu}$ | $\left( \sqrt{k} \frac{\|V\|}{\sqrt{n}} + \sqrt{d} \frac{\|V\|_{1,1}}{n} \right) \left( \frac{\|V\|_F \|V\|_{2,\infty}}{\sqrt{n}} \frac{k^{\frac{3}{2}}}{\varepsilon \nu} \log n + \frac{k^2 d}{\varepsilon} \right)$ |

$\mathcal{D}_{(\|v_i\|)_{i=1}^n}^{(1)}$ used to approximate $\sum_{i \in \mathcal{C}_j^t} v_i$, similar to our classical algorithms. The random variables are basically $X(i,j) = (0^{j-1}, 1, 0^{k-j-1})$ in one case and $X(i,j) = \left( 0^{(j-1)d}, \frac{v_i}{\|v_i\|}, 0^{(k-j-1)d} \right)$ in the other. The outputs of the two multivariate quantum mean estimators are thus good approximations to $|\mathcal{C}_1^t|, \ldots, |\mathcal{C}_k^t|$ and $\sum_{i \in \mathcal{C}_1^t} v_i, \ldots, \sum_{i \in \mathcal{C}_k^t} v_i$. The precise query and time complexities of quantum $(\varepsilon, \nu = 0)$-$k$-means Algorithm 3 are shown in Table 1 below, while its analysis is postponed to Appendix B. Note that Algorithm 3 uses the operations from Items 1 to 4, 7 and 8 of Definition 3.

Mirroring our classical algorithms, we propose a second quantum algorithm (Algorithm 4 in Appendix B) similar to Algorithm 3 but where the distances $\|v_i - c_j^t\|^2$ are approximated up to error $\frac{\nu}{2}$ by using a quantum subroutine (Lemma 18) based on quantum amplitude estimation (Brassard et al., 2002) and quantum variable-time minimum finding (Ambainis, 2012). Another difference from Algorithm 3 is that we create a quantum superposition over the distribution $\mathcal{D}_V^{(1)}$ instead of $\mathcal{D}_{(\|v_i\|)_{i=1}^n}^{(1)}$, i.e., $\sum_{(i,l) \in [n] \times [d]} \sqrt{\frac{|V_{li}|}{\|V\|_{1,1}}} |i, l\rangle$, in order to improve the complexity dependence on $d$. The unitaries behind the states $\sum_{i \in [n]} \frac{1}{\sqrt{n}} |i, \ell_i^t\rangle$ and $\sum_{(i,l) \in [n] \times [d]} \sqrt{\frac{|V_{li}|}{\|V\|_{1,1}}} |i, l, \ell_i^t\rangle$, now with labels $\ell_i^t \in \{j \in [k] : \|v_i - c_j^t\|^2 \le \min_{j' \in [k]} \|v_i - c_{j'}^t\|^2 + \nu\}$, are once again inputted into the multivariate quantum mean estimator from Cornelissen et al. (2022). The precise query and time complexities are described in Table 1, while its analysis is postponed to Appendix B. Note that Algorithm 4 requires the operations from Items 1 to 3 and 5 to 8 of Definition 3.

**Lower bounds.** In Appendix C we prove classical and quantum query lower bounds that show that our algorithms are optimal in most parameters. Our lower bounds come from reducing the problem of approximating the centroids $|\mathcal{C}_j|^{-1} \sum_{i \in \mathcal{C}_j} v_i$ given classical/quantum query access to matrix $V \in \mathbb{R}^{d \times n}$ and classical description of clusters $\{C_j\}_{j \in [k]}$ from the problem of approximating the Hamming weight of some bit-string, whose query complexity is well known (Nayak & Wu, 1999). More specifically, we construct a dataset matrix $V$ for which all points within the same cluster $\mathcal{C}_j$ have the same $\ell_r$-norm for any $r \in [1, \infty]$, so access to $\|v_i\|_r$ does not give any meaningful information about the centroids. An algorithm for approximating $|\mathcal{C}_j|^{-1} \sum_{i \in \mathcal{C}_j} v_i$ would then give an algorithm for approximating $\Theta(kd)$ independent Hamming weights on $\Theta(\frac{n}{k})$ bits each to precision $O\left( \frac{n^{3/2} \varepsilon}{k \|V\|_F} \right)$, for which lower bounds are well known.

Below we provide our lower bounds and a simplified version of our algorithms' query complexity.

**Result 1.** *Let $\varepsilon, \nu > 0$ and $\delta \in (0,1)$. Assume classical/quantum query access to $V \in \mathbb{R}^{d \times n}$. Assume all clusters satisfy $|\mathcal{C}_j^t| = \Omega(\frac{n}{k})$. There are classical and quantum algorithms that out-*

*put centroids consistent with $(\varepsilon, \nu)$-$k$-means with probability $1 - \delta$ and with per-iteration query complexity (up to* polylog *factors in $k$, $d$, $\frac{1}{\varepsilon}$, $\frac{1}{\nu}$, $\frac{1}{\delta}$, $\frac{\|V\|_F}{\sqrt{n}}$)*

$$\textbf{\textit{Classical:}} \ \ \widetilde{O}\bigg( \min\bigg\{ \frac{\|V\|_F^2}{n} \frac{k^2 d}{\varepsilon^2}, \ \bigg( \frac{\|V\|_F^2}{n} + \frac{\|V\|_{1,1}^2}{n^2} \bigg) \frac{\|V\|_F^2 \|V\|_{2,\infty}^2}{n} \frac{k^3}{\varepsilon^2 \nu^2} \bigg\} \bigg);$$

$$\textbf{\textit{Quantum:}} \ \ \widetilde{O}\bigg( \min\bigg\{ \frac{\|V\|_F}{\sqrt{n}} \frac{k^{\frac{3}{2}} d}{\varepsilon} (\sqrt{k} + \sqrt{d}), \ \bigg( \sqrt{k} \frac{\|V\|_F}{\sqrt{n}} + \sqrt{d} \frac{\|V\|_{1,1}}{n} \bigg) \frac{\|V\|_F \|V\|_{2,\infty}}{\sqrt{n}} \frac{k^{\frac{3}{2}}}{\varepsilon \nu} \bigg\} \bigg).$$

*Moreover, with entry-wise query access to $V \in \mathbb{R}^{d \times n}$ and $(\|v_i\|_r)_{i \in [n]}$ for any $r \in [1, \infty]$ and classical description of partition $\{\mathcal{C}_j\}_{j \in [k]}$ of $[n]$, any classical or quantum algorithm that outputs $c_1, \ldots, c_k \in \mathbb{R}^d$ with $\big\| c_j - \frac{1}{|\mathcal{C}_j|} \sum_{i \in \mathcal{C}_j} v_i \big\| \leq \varepsilon$ has query complexity*

$$\textbf{\textit{Classical:}} \ \ \Omega\bigg( \min\bigg\{ \frac{\|V\|_F^2}{n} \frac{kd}{\varepsilon^2}, nd \bigg\} \bigg); \qquad \textbf{\textit{Quantum:}} \ \ \Omega\bigg( \min\bigg\{ \frac{\|V\|_F}{\sqrt{n}} \frac{kd}{\varepsilon}, nd \bigg\} \bigg).$$

## 4 EXPERIMENTAL RESULTS

We conduct numerical experiments to validate the theoretical performance of our proposed classical algorithm, EKMeans (Algorithm 1), against the standard $k$-means algorithm. The experiments are designed to demonstrate the scalability of our approach with respect to the dataset size, $n$, and were conducted in C++ on an Intel® Core™ i5-9300H CPU @ 2.40GHz × 8 using only one core.

All experiments were performed on the MNIST dataset and on synthetic datasets. The MNIST dataset is composed of $60000$ $28 \times 28$ grayscale training images, each pixel described by an integer value in $[0, 255]$. We normalised the pixel values to fall into $[0, 1]$, so each image is described by a vector in $[0, 1]^d$ with $d = 784$. We analysed the performance of EKMeans and the standard $k$-means on subsets of the MNIST dataset of sizes $n \in \{20000, 25000, 30000, 35000, 40000, 45000, 50000\}$ as our main numerical experiment. The subsets were drawn uniformly at random *without* repetition out of the $60000$ training images. For the MNIST dataset we set the number of clusters $k = 10$ and the convergence threshold $\tau = 0.15$. When using EKMeans, we set the approximation parameter $\varepsilon \in \{0.5, 1.0, 1.5, 2.0, 2.5\}$ and the probability parameter $\delta = 0.01$. The sample sizes $p$ and $q$ were calculated dynamically based on the dataset properties as $p = \big\lceil \frac{\|V\|^2}{n} \frac{k^2}{\varepsilon^2} \ln \frac{k}{\delta} \big\rceil$ and $q = \big\lceil \frac{\|V\|_{2,1}^2}{n^2} \frac{k^2}{\varepsilon^2} \ln \frac{k}{\delta} \big\rceil$. As a sense of size, $p \approx 18300$ and $q \approx 6600$ on average for $\varepsilon = 0.5$, while $p \approx 730$ and $q \approx 265$ on average for $\varepsilon = 2.5$. The samples were drawn only once at the beginning of the clustering process rather than at each iteration. The results are averaged over $211$ repetitions with different random seeds for the subset $V$ and centroid initialisation $c_1^0, \ldots, c_k^0$, but keeping the same seed for two executions of EKMeans and standard $k$-means.

Regarding the synthetic datasets, they were created as follows: a number of $k$ auxiliary vectors $u_j \in [-1, 1]^d$ were uniformly sampled entry-wise, and for each $j \in [k]$, a number of $\frac{n}{k}$ dataset vectors were obtained as $v_i = u_j + w_i$, where each $w_i \in [-1, 1]^d$ is a vector with uniformly random entries in $[-1, 1]$. In summary, the vectors $v_i$ were uniformly sampled around $k$ uniformly sampled auxiliary centers $u_j$, thus creating $k$ clusters of dataset vectors on average. We analysed the performance of EKMeans and the standard $k$-means on varying synthetic dataset of sizes $n \in \{100000, 150000, 200000, 250000, 300000, 350000, 400000, 450000, 500000\}$. For synthetic datasets we set the number of clusters $k = 5$, the dimension $d = 30$, and the convergence threshold $\tau = 0.1$. When using EKMeans, we set the approximation parameter $\varepsilon \in \{0.2, 0.4, 0.6, 0.8, 1.0\}$ and the probability parameter $\delta = 0.01$. The sample sizes $p$ and $q$ were calculated dynamically based on the dataset properties as $p = \big\lceil \frac{\|V\|^2}{n} \frac{k^2}{\varepsilon^2} \ln \frac{k}{\delta} \big\rceil$ and $q = \big\lceil \frac{\|V\|_{2,1}^2}{n^2} \frac{k^2}{\varepsilon^2} \ln \frac{k}{\delta} \big\rceil$. As a sense of size, for $\varepsilon = 0.2$, $p \approx 20000$ and $q \approx 128000$ on average, while for $\varepsilon = 1.0$, $p \approx 800$ and $q \approx 5100$ on average. The samples were drawn only once at the beginning of the clustering process. The results are averaged over $4325$ repetitions with different random seeds for dataset $V$ and centroid initialisation $c_1^0, \ldots, c_k^0$, but keeping the same seed for two executions of EKMeans and standard $k$-means.

Figure 1a (MNIST) and Figure 2a (synthetic) show the total runtime required to compute centroids $c_1^t, \ldots, c_k^t$ which satisfy the convergence criteria $\frac{1}{k} \sum_{j \in [k]} \|c_j^t - c_j^{t-1}\| \leq \tau$. This includes sampling the sets $P, Q \subseteq [n]$. As predicted by the theoretical complexity $O(nkd)$, the total runtime for

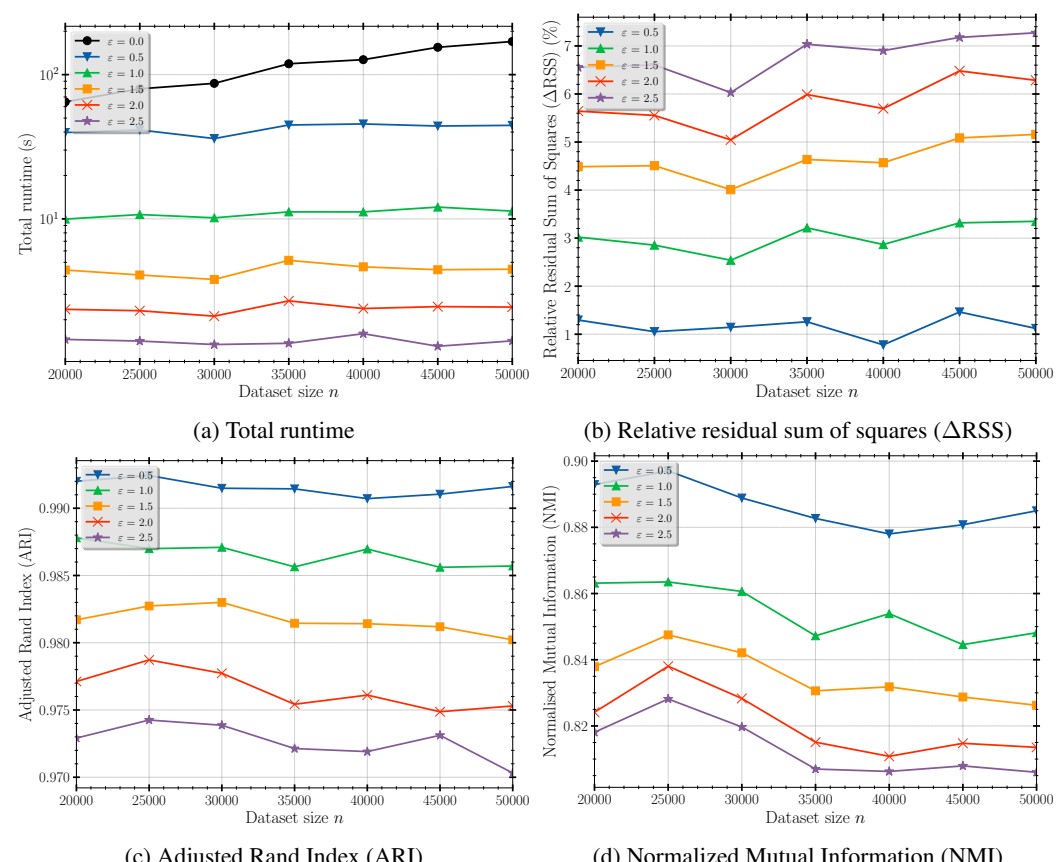

Figure 1: Total runtime, relative residual sum of squares ($\Delta\mathrm{RSS}$), adjusted Rand index (ARI), and normalized mutual information (NMI) as a function of the MNIST dataset size $n$. Here $k = 10$, $d = 784$, $\delta = 0.01$, and $\tau = 0.15$. The standard $k$-means is depicted as $\varepsilon = 0$. Each point is the average of 211 random datasets and centroid initialisations.

standard $k$-means grows linearly with the dataset size $n$. In contrast, the total runtime for EKMeans remains nearly constant across different $n$'s (a more thorough analysis on the dependence on $n$ is left to Appendix D). This empirically validates that time complexity of our algorithm barely scales with $n$, a massive improvement over the standard approach. On synthetic datasets with $n = 500000$, $k$-means requires $\approx 9$ s on average to run, while EKMeans with $\varepsilon = 1.0$ requires only $\approx 45$ ms on average to run, a 200-fold improvement! On the other hand, on the MNIST dataset with $n = 50000$, $k$-means requires $\approx 171$ s on average to run, while EKMeans with $\varepsilon = 2.5$ requires only $\approx 1.4$ s on average to run, a 120-fold improvement.

In Figure 1b (MNIST) and Figure 2b (synthetic), we compare the accuracy performance of EKMeans and of $k$-means measured by their residual sum of squares $\mathrm{RSS} := \sum_{i \in [n]} \min_{j \in [k]} \|v_i - c_j^t\|$. More precisely, we analyse the relative difference $(\mathrm{RSS}_\varepsilon - \mathrm{RSS}_0)/\mathrm{RSS}_0$ between the $\mathrm{RSS}_\varepsilon$ of EKMeans with approximation parameter $\varepsilon$ and the $\mathrm{RSS}_0$ of $k$-means, which we call *relative residual sum of squares* ($\Delta\mathrm{RSS}$). The performance of EKMeans on synthetic datasets is only around $0.5\%$ worse relative to $k$-means, even at larger values of $\varepsilon$. On the MNIST dataset, the performance of EKMeans is around $7\%$ worse for $\varepsilon = 2.5$, a relatively small deviation that is more than outweighed by the faster runtimes. Such a difference is a consequence of the MNIST dataset not being as well clustered as our synthetic datasets.

In Figure 1c (MNIST) and Figure 2c (synthetic), we compare the performance of EKMeans and of $k$-means via the Adjusted Rand Index (ARI), while in Figure 1d (MNIST) and Figure 2d (synthetic), we compare the algorithms via the Normalised Mutual Information (NMI). Given clusterings $\{\mathcal{C}_j^{t,0}\}_{j \in [k]}$ and $\{\mathcal{C}_j^{t,\varepsilon}\}_{j \in [k]}$ due to $k$-means and EKMeans with parameter $\varepsilon$, respectively, let

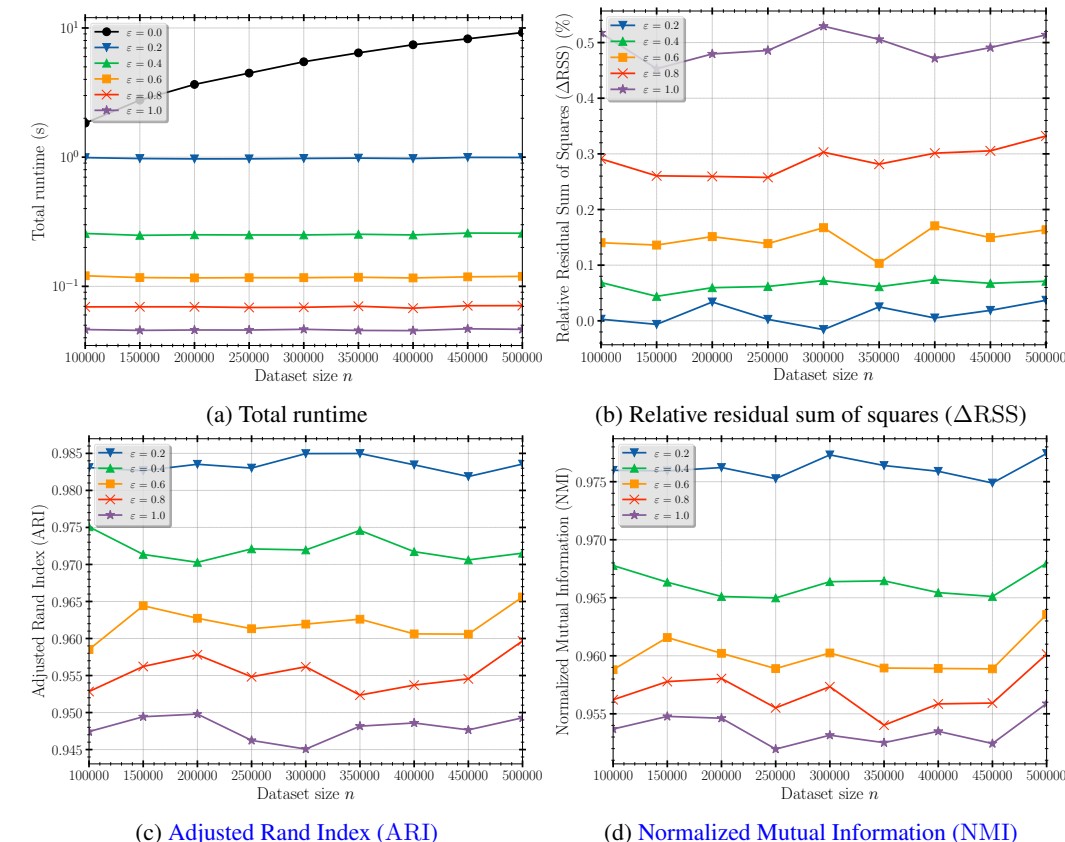

(a) Total runtime

(b) Relative residual sum of squares ($\Delta$RSS)

(c) Adjusted Rand Index (ARI)

(d) Normalized Mutual Information (NMI)

Figure 2: Total runtime, relative residual sum of squares ($\Delta$RSS), adjusted Rand index (ARI), and normalized mutual information (NMI) as a function of the synthetic dataset size $n$. Here $k = 5$, $d = 30$, $\delta = 0.01$, and $\tau = 0.1$. The standard $k$-means is depicted as $\varepsilon = 0$. Each point is the average of 4325 random datasets and centroid initialisations.

$N_{ij} = |\mathcal{C}_i^{t,0} \cap \mathcal{C}_j^{t,\varepsilon}|$, $a_i = \sum_{j \in [n]} N_{ij}$, and $b_j = \sum_{i \in [n]} N_{ij}$ for $i, j \in [n]$. Then

$$\text{ARI} := \frac{\sum_{i,j \in [n]} \binom{N_{ij}}{2} - \sum_{i,j \in [n]} \binom{a_i}{2}\binom{b_j}{2} / \binom{n}{2}}{\frac{1}{2} \sum_{i \in [n]} \binom{a_i}{2} + \frac{1}{2} \sum_{j \in [n]} \binom{b_j}{2} - \sum_{i,j \in [n]} \binom{a_i}{2}\binom{b_j}{2} / \binom{n}{2}},$$

$$\text{NMI} := -\frac{\sum_{i,j \in [n]} \frac{N_{ij}}{n} \log\left(\frac{N_{ij}/n}{(a_i/n)(b_j/n)}\right)}{\frac{1}{2} \sum_{i \in [n]} \frac{a_i}{n} \log \frac{a_i}{n} + \frac{1}{2} \sum_{j \in [n]} \frac{b_j}{n} \log \frac{b_j}{n}}.$$

According to both measures, the final clusterings via $k$-means and EKMeans are similar. The ARI is very close to 1 for all values of $\varepsilon$, e.g., around $0.97$ for $\varepsilon = 2.5$ on the MNIST dataset and $0.95$ for $\varepsilon = 1.0$ on the synthetic dataset. The NMI, on the other hand, is slightly worse on the MNIST dataset but still sufficiently large, at around $0.8$ in the worst case, while on the synthetic dataset it is larger than $0.95$ for all considered values of $\varepsilon$. Finally, we mention that further numerical experiments are conducted in Appendix D.

## 5 RELATED INDEPENDENT WORK

We briefly mention related works that have appeared online around the same time or later than ours. First, the independent work of Jaiswal (2023) (see also Shah & Jaiswal (2025)) quantised the highly parallel, sampling-based approximation scheme of Bhattacharya et al. (2020) and thus obtained a quantum algorithm for the $k$-means problem with *provable* guarantees, as opposed to our results, which are heuristic. Due to such a guarantee, though, their final runtime depends exponentially

in $k$ and $\frac{1}{\varepsilon}$ (but maintains the polylogarithmic dependence on $n$). Another related work is Xue et al. (2023), who proposed a quantum algorithm to compute coresets for $k$-means, a compressed representation of the dataset which preserves the optimal residual sum of squares up to some small multiplicative error $\varepsilon$. By employing $\widetilde{O}(\sqrt{nk}d^{\frac{3}{2}}/\varepsilon)$ QRAM calls, the authors obtained a coreset of size $O\left(\frac{kd}{\varepsilon^2}\operatorname{poly}\log n\right)$. Their coreset can then be used by classical algorithms to obtain provable guarantees. Nonetheless, our query complexities are independent on $n$, far better than Xue et al. (2023). Finally, Poggiali et al. (2024) proposed hybrid quantum-classical $k$-means algorithms that reduce the complexity of the cluster assignment step by a constant cost.

Very recently, Chen et al. (2025) proposed alternative quantum algorithms to the ones presented here by employing uniform sampling plus shifting the vectors $v_i$ by the current centroids $c_1^t, \dots, c_k^t$. Their algorithm uses $\widetilde{O}\left(k^{\frac{5}{2}}\sqrt{d}\left(\frac{\sqrt{\phi}}{\varepsilon} + \sqrt{d}\right)\right)$ QRAM calls, where $\phi := \frac{1}{n}\sum_{j\in[k]}\sum_{i\in\mathcal{C}_j^t}\|v_i - c_j^{t+1}\|^2$ and $c_j^{t+1} = |\mathcal{C}_j^t|^{-1}\sum_{i\in\mathcal{C}_j^t} v_i$ for clusters $\{\mathcal{C}_j^t\}_{j\in[k]}$ defined by $c_1^t, \dots, c_k^t$. Since one can write $\phi = \frac{1}{n}\sum_{j\in[k]}\left(\sum_{i\in\mathcal{C}_j^t}\|v_i\|^2 - |\mathcal{C}_j^t|^{-1}\|\sum_{i\in\mathcal{C}_j^t} v_i\|^2\right)$, then $\sqrt{\phi} \le \frac{\|V\|_F}{\sqrt{n}}$. Algorithm 3, on the other hand, makes $\widetilde{O}\left(\left(\sqrt{k}\frac{\|V\|}{\sqrt{n}} + \sqrt{d}\frac{\|V\|_{2,1}}{n}\right)\frac{k^{3/2}}{\varepsilon}\right)$ QRAM calls assuming $|i\rangle|\bar{0}\rangle \mapsto |i\rangle|v_i\rangle$ counts as 1 QRAM call as in Chen et al. (2025) (Definition 3 assumes that this map counts as $d$ QRAM calls instead). If $\sqrt{\phi} \ll \frac{\|V\|}{\sqrt{n}} + \frac{\|V\|_{2,1}}{n}$ (note that $\|V\| \le \|V\|_F$ and $\frac{\|V\|_{2,1}}{n} \le \frac{\|V\|_F}{\sqrt{n}}$), e.g., when the distance between vectors within the same cluster is much smaller than the distance between clusters, the complexity from Chen et al. (2025) can be better than ours, otherwise, if $\sqrt{\phi} = \omega\left(\frac{1}{\sqrt{kd}}\frac{\|V\|}{\sqrt{n}} + \frac{1}{k}\frac{\|V\|_{2,1}}{n}\right)$, our complexity is better. Both results are thus incomparable.

# 6 CONCLUSIONS AND FUTURE DIRECTIONS

We proposed improved classical and quantum approximation versions of the standard $k$-means algorithm with runtimes depending only logarithmically on the size $n$ of the dataset $V$. Our algorithms not only match the dependence on $n$ from the quantum $q$-means algorithm of Kerenidis et al. (2019) but also improve the dependence on several other parameters like number of clusters $k$, dimension $d$, approximation parameter $\varepsilon$, and other parameters depending on $V$. For such, we assumed that the dataset $V$ has been pre-processed beforehand to allow for efficient sampling and query operations. The required data structures have been previously used in other (quantum) machine learning applications (Hazan et al., 2011; Song et al., 2016; Kerenidis & Prakash, 2017; Biamonte et al., 2017; Tang, 2019). We note that our classical algorithms can be seen as a "dequantised" version of our quantum algorithms, in a similar flavor to prior dequantisation works (Tang, 2019; 2021; Gilyén et al., 2018; 2022). Moreover, our upper bounds were complemented with query lower bounds, proving that our algorithms are optimal in several parameters, which hints at our choice for subroutines being right.

Even though our quantum algorithms require the use of QRAM, we are not aware of any inherent reason why this model would not be physically realisable in the lab. Indeed, several new results suggest otherwise (Hann et al., 2021; Mehta et al., 2024; Dalzell et al., 2025; Shen et al., 2025). Nonetheless, we do believe that designing quantum algorithms in the QRAM-model, like in this work, can help motivate the development of such architectures in the lab, and inform their role in the algorithmic frameworks they are to be embedded in.

Finally, we conducted numerical experiments to measure the performance of our main classical algorithm, EKMeans, compared to the standard $k$-means. Our findings support our theoretical results in that EKMeans has time complexity *almost independent* on the dataset size $n$ while still returning centroids on par with $k$-means quality-wise. Even more impressive, EKMeans is extremely fast, running at the order of tens of milliseconds for datasets reaching the size of millions! We believe that EKMeans can become a competitive clustering algorithm, specially if a given dataset must be analysed repeated times so that pre-processing it makes sense. This is probably one of the first examples of a practically viable dequantised algorithm.

We mention a few future directions. One is to assume data vectors with special properties, e.g., well-clusterable datasets (Kerenidis et al., 2019), in order to obtain tighter runtimes. In this direction, Chen et al. (2025) exploited certain symmetries of $k$-means. Another direction is to bridge our work and Jaiswal (2023) to obtain improved complexities in $k$ and $\frac{1}{\varepsilon}$ together with provable guarantees.

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

## A    CLASSICAL ALGORITHMS

We now present our classical $(\varepsilon, \nu)$-$k$-means algorithms, whose main idea is to employ the classical query access from Definition 2 to separately estimate the quantities $\sum_{i \in \mathcal{C}_j^t} v_i$ and $|\mathcal{C}_j^t|$ for $j \in [k]$, from which the new centroids $c_j^{t+1} \approx |\mathcal{C}_j^t|^{-1} \sum_{i \in \mathcal{C}_j^t} v_i$ are approximated. For our first algorithm, we sample columns of $V$ with probability $\frac{\|v_i\|}{\|V\|_{2,1}}$ and select those closer to $c_j^t$ to approximate $\sum_{i \in \mathcal{C}_j^t} v_i$. To approximate $|\mathcal{C}_j^t|$ we sample columns of $V$ uniformly at random instead.

Before presenting and proving the correctness of our classical algorithm, we recall the following useful concentration inequalities and approximation lemma.

**Fact 4** (Chernoff's bound). *Let $X := \sum_{i \in [N]} X_i$ where $X_1, \ldots, X_N$ are independently distributed in $[0, 1]$. Then $\Pr[|X - \mathbb{E}[X]| \geq \epsilon \mathbb{E}[X]] \leq 2e^{-\epsilon^2 \mathbb{E}[X]/3}$ for all $\epsilon > 0$.*

**Fact 5** (Median-of-means ([Lerasle, 2019](#), Proposition 12)). *Let $X^{(j)} := \frac{1}{N} \sum_{i \in [N]} X_i^{(j)}$ for $j \in [K]$, where $\{X_i^{(j)}\}_{i \in [N], j \in [K]}$ are i.i.d. copies of a random variable $X$. Then, for all $\epsilon > 0$ and $\sigma^2 \geq \mathrm{Var}(X)$,*

$$\Pr[|\mathrm{median}(X^{(1)}, \ldots, X^{(K)}) - \mathbb{E}[X]| \geq \epsilon] \leq \exp\left(-2K\left(\frac{1}{2} - \frac{\sigma^2}{N\epsilon^2}\right)^2\right).$$

**Fact 6** (Freedman's inequality ([Freedman, 1975](#), Theorem 1.6) & ([Tropp, 2011](#), Theorem 1.1)). *Let $\{Y_i : i \in \mathbb{N} \cup \{0\}\}$ be a real-valued martingale with difference sequence $\{X_i : i \in \mathbb{N}\}$. Assume that $X_i \leq B$ almost surely for all $i \in \mathbb{N}$. Let $W_i := \sum_{j \in [i]} \mathbb{E}[X_j^2 | X_1, \ldots, X_{j-1}]$ for $i \in \mathbb{N}$. Then, for all $\epsilon \geq 0$ and $\sigma > 0$,*

$$\Pr[\exists i \geq 0 : Y_i \geq \epsilon \text{ and } W_i \leq \sigma^2] \leq \exp\left(-\frac{\epsilon^2/2}{\sigma^2 + B\epsilon/3}\right).$$

**Claim 7.** *Let $\widetilde{a}, a \in \mathbb{R}_+$ be such that $|a - \widetilde{a}| \leq \epsilon$, where $\epsilon \in [0, \frac{a}{2}]$. Then $\left|\frac{1}{\widetilde{a}} - \frac{1}{a}\right| \leq \frac{2\epsilon}{a^2}$.*

**Theorem 8** (Classical $(\varepsilon, 0)$-$k$-means algorithm). *Let $\varepsilon > 0$, $\delta \in (0, 1)$, and assume classical query access to $V = [v_1, \ldots, v_n] \in \mathbb{R}^{d \times n}$. If all clusters satisfy $|\mathcal{C}_j^t| = \Omega(\frac{n}{k})$, then [Algorithm 1](#) outputs centroids consistent with the $(\varepsilon, \nu = 0)$-$k$-means algorithm with probability $1 - \delta$. The complexities per iteration of [Algorithm 1](#) are*

$$\textit{Classical queries: } O\left(\left(\frac{\|V\|^2}{n} + \frac{\|V\|_{2,1}^2}{n^2}\right)\frac{k^2 d}{\varepsilon^2} \log \frac{k}{\delta}\right),$$

$$\textit{Time: } O\left(\left(\frac{\|V\|^2}{n} + \frac{\|V\|_{2,1}^2}{n^2}\right)\frac{k^2 d}{\varepsilon^2}(k + \log(nd)) \log \frac{k}{\delta}\right).$$

*Proof.* Let $\chi_j^t \in \mathbb{R}^n$ be the characteristic vector for cluster $j \in [k]$ at iteration $t$ scaled to $\ell_1$-norm, i.e., $(\chi_j^t)_i = \frac{1}{|\mathcal{C}_j^t|}$ if $i \in \mathcal{C}_j^t$ and $0$ if $i \notin \mathcal{C}_j^t$. For $i \in [n]$, let $\ell_i^t := \arg\min_{j \in [k]} \|v_i - c_j^t\|$. Sample $p$ indices $P \subseteq [n]$ uniformly from $[n]$. Let $\lambda_j = \frac{p}{|P_j|} \frac{|\mathcal{C}_j^t|}{n}$, where $P_j := \{i \in P | \ell_i^t = j\}$. On the other hand, sample $q$ indices $Q \subseteq [n]$ from the distribution $\mathcal{D}_{(\|v_i\|)_{i=1}^n}^{(1)}(i) = \frac{\|v_i\|}{\|V\|_{2,1}}$. For each $i \in Q$, let $X_i \in \mathbb{R}^{d \times n}$ be the matrix formed by setting the $i$-th column of $X_i$ to $\|V\|_{2,1} \frac{v_i}{\|v_i\|}$ and the rest to zero. Define $\widetilde{V} := \frac{1}{q} \sum_{i \in Q} X_i$. Then $\mathbb{E}[\widetilde{V}] = V$.

We start with the error analysis. We note that the outputs of the standard $k$-means and [Algorithm 1](#) can be stated, respectively, as $c_j^{* \, t+1} = V \chi_j^t$ and $c_j^{t+1} = \lambda_j \widetilde{V} \chi_j^t = \frac{p}{n|P_j|} \sum_{i \in Q_j} \frac{\|V\|_{2,1}}{q} \frac{v_i}{\|v_i\|}$, where $Q_j := \{i \in Q | \ell_i^t = j\}$. In order to bound $\|c_j^{* \, t+1} - c_j^{t+1}\|$, first note that, by the triangle inequality,

$$\|c_j^{* \, t+1} - c_j^{t+1}\| \leq |\lambda_j - 1| \|V \chi_j^t\| + |\lambda_j| \|(\widetilde{V} - V)\chi_j^t\|,$$

so we aim at bounding $|\lambda_j - 1| \leq \frac{\varepsilon}{2\|V\chi_j^t\|}$ and $\|(\widetilde{V} - V)\chi_j^t\| \leq \frac{\varepsilon}{2|\lambda_j|}$. Let us start with $|\lambda_j - 1|$. Notice that $|P_j|$ is a binomial random variable with mean $p|\mathcal{C}_j^t|/n$. By a Chernoff bound ([Fact 4](#)),

$$\Pr\left[\left|\frac{|P_j|}{p}\frac{n}{|\mathcal{C}_j^t|} - 1\right| \geq \frac{\varepsilon}{4\|V\|\|\chi_j^t\|}\right] \leq 2\exp\left(-\frac{\varepsilon^2 p|\mathcal{C}_j^t|}{48n\|V\|^2\|\chi_j^t\|^2}\right). \tag{1}$$

It suffices to take $p = \frac{48\|V\|^2\|\chi_j^t\|^2 n}{\varepsilon^2 |\mathcal{C}_j^t|} \ln \frac{2k}{\delta} = O\left(\frac{\|V\|^2}{n}\frac{k^2}{\varepsilon^2} \log \frac{k}{\delta}\right)$ in order to estimate $|\lambda_j^{-1} - 1| \leq \frac{\varepsilon}{4\|V\|\|\chi_j^t\|}$ with probability at least $1 - \frac{\delta}{2k}$ (using that $|\mathcal{C}_j^t| = \Omega(\frac{n}{k})$, $\|V\chi_j^t\| \leq \|V\|\|\chi_j^t\|$, and $\|\chi_j^t\|^2 = 1/|\mathcal{C}_j^t|$). The bound on $|\lambda_j^{-1} - 1|$ implies that $|\lambda_j - 1| \leq \frac{\varepsilon}{2\|V\|\|\chi_j^t\|}$, where we used [Claim 7](#)

and that $|\lambda_j| \leq 2$ with high probability — which is already implied by the bound in Eq. (1). By the union bound, the bound on $\lambda_j$ holds for all $j \in [k]$ with probability at least $1 - \frac{\delta}{2}$.

Regarding the bound on $\widetilde{V}$, we use Freedman's inequality to prove $\|(\widetilde{V} - V)\chi_j^t\| \leq \frac{\varepsilon}{4} \leq \frac{\varepsilon}{2|\lambda_j|}$ (again using that $|\lambda_j| \leq 2$). For such, let $f(X_1, \ldots, X_q) = \|(\widetilde{V} - V)\chi_j^t\|$. First, for all $i \in [q]$,

$$
\begin{aligned}
|f(X_1, \ldots, X_i, \ldots, X_q) - f(X_1, \ldots, X_i', \ldots, X_q)| &= \left| \|(\widetilde{V} - V)\chi_j^t\| - \|(\widetilde{V}' - V)\chi_j^t\| \right| \\
&\leq \|(\widetilde{V} - \widetilde{V}')\chi_j^t\| = \frac{1}{q}\|(X_i - X_i')\chi_j^t\| \leq \frac{2}{q}\|X_i\chi_j^t\| \leq \frac{2\|V\|_{2,1}}{q|\mathcal{C}_j^t|}.
\end{aligned}
$$

Second, we bound the variance: for all $i \in [q]$,

$$
\mathbb{E}_{X_i, X_i'}\left[ f(X_1, \ldots, X_i, \ldots, X_q) - f(X_1, \ldots, X_i', \ldots, X_q) \right]^2 \leq \frac{1}{q^2}\mathbb{E}_{X_i, X_i'}\left[ \|(X_i - X_i')\chi_j^t\| \right]^2
$$

$$
\leq \frac{4}{q^2}\mathbb{E}_{X_i}[\|X_i\chi_j^t\|^2] = \frac{4}{q^2}(\chi_j^t)^\top \mathbb{E}_X[X^\top X]\chi_j^t = \frac{4}{q^2}(\chi_j^t)^\top \|V\|_{2,1}\operatorname{diag}((\|v_i\|)_{i \in [n]})\chi_j^t \leq \frac{4\|V\|_{2,1}^2}{q^2|\mathcal{C}_j^t|^2}.
$$

We employ Freedman's inequality with the Doob martingale $Y_i := \mathbb{E}[f(X_1, \ldots, X_q)|X_1, \ldots, X_i]$ for $i \in [q]$. Then Fact 6 with $B = \frac{2\|V\|_{2,1}}{q|\mathcal{C}_j^t|}$ and $\sigma^2 = q \cdot \frac{4\|V\|_{2,1}^2}{q^2|\mathcal{C}_j^t|^2}$ leads to

$$
\Pr\left[ \|(\widetilde{V} - V)\chi_j^t\| \geq \frac{\varepsilon}{4} \right] \leq \exp\left( -\frac{\varepsilon^2/32}{\frac{4\|V\|_{2,1}^2}{q|\mathcal{C}_j^t|^2} + \frac{\varepsilon\|V\|_{2,1}}{6q|\mathcal{C}_j^t|}} \right).
$$

It suffices to take $q = O\left(\max\left\{\frac{\|V\|_{2,1}^2}{n^2}\frac{k^2}{\varepsilon^2}, \frac{\|V\|_{2,1}}{n}\frac{k}{\varepsilon}\right\}\log\frac{k}{\delta}\right)$ to approximate $\|(\widetilde{V} - V)\chi_j^t\| \leq \frac{\varepsilon}{2}$ with probability at least $1 - \frac{\delta}{2k}$ (already using that $|\mathcal{C}_j^t| = \Omega(\frac{n}{k})$). All in all, we have $\|c_j^{* \, t+1} - c_j^{t+1}\| \leq \varepsilon$ with probability at least $1 - \delta$ for all the centroids (using a union bound).

We now turn our attention to the query and time complexities. In order to compute the clusters $\{\mathcal{C}_j^t\}_{j \in [k]}$, for each $i \in P \cup Q$ and $j \in [k]$, we exactly compute the distance $\|v_i - c_j^t\|$, which requires $O(d\log(nd))$ time: $O(d\log(nd))$ time to read a vector of $d$ components $v_i$ and $O(d)$ time to compute the distance. In total, we need access to at most $p + q$ vectors, so $O((p+q)d)$ classical queries, while the time complexity is $O((p+q)(kd + d\log(nd)))$, accounting for the $O(kd)$ cost in computing all distances $\|v_i - c_j^t\|$ between $v_i$ and the $k$ centroids stored in memory. Finally, the $k$ new centroids are obtained by summing $q$ $d$-dimensional vectors in $O(qd)$ time. In summary,

- Sampling $P, Q \subseteq [n]$ and querying the corresponding vectors $\{v_i\}_{i \in P \cup Q}$ takes $O((p+q)d)$ queries and $O((p+q)d\log(nd))$ time;

- Obtaining the labels $\{\ell_i^t\}_{i \in P \cup Q}$ requires $O((p+q)kd)$ time;

- Computing new centroids $\{c_j^{t+1}\}_{j \in [k]}$ by adding $q$ $d$-dimensional vectors takes $O(qd)$ time.

This means the overall query complexity is $O((p+q)d)$ and the time complexity is

$$
O\big((p+q)d(k + \log(nd))\big) = O\left( \left( \frac{\|V\|^2}{n} + \frac{\|V\|_{2,1}^2}{n^2} \right) \frac{k^2 d}{\varepsilon^2}(k + \log(nd))\log\frac{k}{\delta} \right). \qquad \square
$$

Using the same techniques and observations, we can also offer guarantees on the approximation error of a single iteration of mini-batch $k$-means algorithms. The difference between mini-batch $k$-means and Algorithm 1 consists in the probability distribution one samples from in order to approximate $\sum_{i \in \mathcal{C}_j^t} v_i$: the former samples $Q \subseteq [n]$ uniformly at random.

**Theorem 9** (Mini-batch $k$-means algorithm). *Let $\varepsilon > 0$ and $\delta \in (0, 1)$. If all clusters satisfy $|\mathcal{C}_j^t| = \Omega(\frac{n}{k})$, then mini-batch $k$-means outputs centroids consistent with the $(\varepsilon, \nu = 0)$-$k$-means*

---

**Algorithm 2** Classical $(\varepsilon, \nu)$-$k$-means algorithm

---

**Input:** Classical query access to data matrix $V = [v_1, \ldots, v_n] \in \mathbb{R}^{d \times n}$, parameters $\delta, \varepsilon, \nu$.
1: Select $k$ initial centroids $c_1^0, \ldots, c_k^0$
2: **for** $t = 0$ until convergence **do**
3:  Sample $p = O\left(\frac{\|V\|^2}{n} \frac{k^2}{\varepsilon^2} \log \frac{k}{\delta}\right)$ rows $P \subseteq [n]$ uniformly from $[n]$
4:  Sample $q = O\left(\frac{\|V\|_{1,1}^2}{n^2} \frac{k^2}{\varepsilon^2} \log \frac{k}{\delta}\right)$ rows $Q \subseteq [n] \times [d]$ from $\mathcal{D}_V^{(1)}$
5:  For $i \in P$ and $(i, \cdot) \in Q$, compute $\ell_i^t \in \{j \in [k] : \|v_i - c_j^t\|^2 \le \min_{j' \in [k]} \|v_i - c_{j'}^t\|^2 + \nu\}$
   using Lemma 11
6:  For $(j, l) \in [k] \times [d]$, let $P_j := \{i \in P | \ell_i = j\}$ and $Q_{jl} := \{(i, l') \in Q | (\ell_i^t, l') = (j, l)\}$
7:  For $(j, l) \in [k] \times [d]$, let the new centroids $(c_j^{t+1})_l = \frac{p}{n|P_j|} \sum_{(i,l') \in Q_{jl}} \frac{\|V\|_{1,1}}{q} \operatorname{sgn}(V_{l'i})$
8: **end for**

---

*algorithm with probability $1 - \delta$. The complexities per iteration are*

$$\textbf{\textit{Classical queries:}} \; O\left(\frac{\|V\|_F^2}{n} \frac{k^2 d}{\varepsilon^2} \log \frac{k}{\delta}\right),$$

$$\textbf{\textit{Time:}} \; O\left(\frac{\|V\|_F^2}{n} \frac{k^2 d}{\varepsilon^2} (k + \log(nd)) \log \frac{k}{\delta}\right).$$

*Proof.* The proof is very similar to Theorem 8, the main difference being that we now sample $q$ indices $Q \subseteq [n]$ uniformly at random and the matrix $X_i \in \mathbb{R}^{d \times n}$ is now formed by setting the $i$-th column of $X_i$ to $nv_i$ and the rest to zero. Hence, the matrix $\widetilde{V}$ is defined as $\widetilde{V} := \frac{1}{q} \sum_{i \in Q} X_i$, which in expectation (under the uniform distribution) is equal to $V$. The new centroids are defined as $c_j^{t+1} = \lambda_j \widetilde{V} \chi_j^t = \frac{p}{|P_j|} \frac{1}{q} \sum_{i \in Q_j} v_i$. Similarly to Theorem 8, it suffices to take $p = \frac{48 \|V\|^2 \|\chi_j^t\|^2 n}{\varepsilon^2 |\mathcal{C}_j^t|} \ln \frac{2k}{\delta} = O\left(\frac{\|V\|^2}{n} \frac{k^2}{\varepsilon^2} \log \frac{k}{\delta}\right)$ in order to estimate $|\lambda_j - 1| \le \frac{\varepsilon}{2\|V\|\|\chi_j^t\|}$ for all $j \in [k]$ with probability at least $1 - \frac{\delta}{2}$.

On the other hand, in order to bound $\|(\widetilde{V} - V)\chi_j^t\| \le \frac{\varepsilon}{4} \le \frac{\varepsilon}{2|\lambda_j|}$, we once again employ Freedman's inequality, now with range and variance bounded by

$$|f(X_1, \ldots, X_i, \ldots, X_q) - f(X_1, \ldots, X_i', \ldots, X_q)| \le \frac{2n\|V\|_{2,\infty}}{q|\mathcal{C}_j^t|},$$

$$\mathbb{E}_{X_i, X_i'} \left[f(X_1, \ldots, X_i, \ldots, X_q) - f(X_1, \ldots, X_i', \ldots, X_q)\right]^2 \le \frac{4n\|V\|_F^2}{q^2|\mathcal{C}_j^t|^2}.$$

Freedman's inequality thus yields that $\|(\widetilde{V} - V)\chi_j^t\| \le \frac{\varepsilon}{4}$ for all $j \in [k]$ with probability at least $1 - \frac{\delta}{2}$ if we take $q = O\left(\max\left\{\frac{\|V\|_F^2}{n} \frac{k^2}{\varepsilon^2}, \frac{k\|V\|_{2,\infty}}{\varepsilon}\right\} \log \frac{k}{\delta}\right)$. Hence, the total number of queries is $O\left(\frac{\|V\|_F^2}{n} \frac{k^2 d}{\varepsilon^2} \log \frac{k}{\delta}\right)$. The runtime of the algorithm follows analogously. $\square$

Since $\frac{\|V\|_{2,1}^2}{n^2} \le \frac{\|V\|_F^2}{n}$ and $\|V\| \le \|V\|_F$, it is easy to see that Algorithm 1 has better sample and time complexities than a mini-batch $k$-means algorithm.

Algorithm 1 computes the distances $\|v_i - c_j^t\|$, and thus the labels $\{\ell_i^t\}_{i \in P \cup Q}$, in an exact way via classical arithmetic circuits in $O((p + q)kd)$ time ($O(d)$ time for each pair $(v_i, c_j^t)$). Similarly, the new centroids $c_j^{t+1}$ are computed by adding $q$ $d$-dimensional vectors, $\sum_{i \in Q_j} \frac{v_i}{\|v_i\|}$. It is possible, however, to approximate $\|v_i - c_j^t\|$ via a sampling procedure, which allows to trade the dependence on $d$ with some norm of $V$. Algorithm 2 describes how this can be performed and the next theorem analyses its query and time complexities.

**Theorem 10** (Classical $(\varepsilon, \nu)$-$k$-means algorithm)**.** *Let $\varepsilon, \nu > 0$, $\delta \in (0, 1)$, and assume classical query access to $V = [v_1, \ldots, v_n] \in \mathbb{R}^{d \times n}$. If all clusters satisfy $|\mathcal{C}_j^t| = \Omega(\frac{n}{k})$, then Algorithm 2 outputs centroids consistent with the $(\varepsilon, \nu)$-$k$-means algorithm with probability $1 - \delta$. The complexities*

*per iteration of* Algorithm 2 *are (up to* polylog *factors in* $k$, $d$, $\frac{1}{\delta}$, $\frac{1}{\nu}$, $\frac{1}{\varepsilon}$, $\frac{\|V\|_F}{\sqrt{n}}$)

$$\textbf{\textit{Classical queries:}} \ \widetilde{O}\bigg(\bigg(\frac{\|V\|^2}{n} + \frac{\|V\|_{1,1}^2}{n^2}\bigg)\frac{\|V\|_F^2\|V\|_{2,\infty}^2}{n}\frac{k^3}{\varepsilon^2\nu^2}\bigg),$$

$$\textbf{\textit{Time:}} \ \widetilde{O}\bigg(\bigg(\frac{\|V\|^2}{n} + \frac{\|V\|_{1,1}^2}{n^2}\bigg)\frac{\|V\|_F^2\|V\|_{2,\infty}^2}{n}\frac{k^3}{\varepsilon^2\nu^2}\log n\bigg).$$

*Proof.* The proof is similar to Theorem 8. Let $\chi_j^t \in \mathbb{R}^n$ be such that $(\chi_j^t)_i = \frac{1}{|\mathcal{C}_j^t|}$ if $i \in \mathcal{C}_j^t$ and 0 if $i \notin \mathcal{C}_j^t$. For $i \in [n]$, let $\ell_i^t \in \{j \in [k] : \|v_i - c_j^t\|^2 \leq \min_{j' \in [k]} \|v_i - c_{j'}^t\|^2 + \nu\}$. Again we sample $p$ indices $P \subseteq [n]$ uniformly from $[n]$ and let $\lambda_j = \frac{p}{|P_j|}\frac{|\mathcal{C}_j^t|}{n}$, where $P_j := \{i \in P | \ell_i^t = j\}$. On the other hand, we now sample $q$ indices $Q \subseteq [n] \times [d]$ from the distribution $\mathcal{D}_V^{(1)}(i,l) = \frac{|V_{li}|}{\|V\|_{1,1}}$. For each $(i,l) \in Q$, let $X_{li} \in \mathbb{R}^{d \times n}$ be the matrix formed by setting the $(l,i)$-th entry of $X_{li}$ to $\|V\|_{1,1}\,\mathrm{sgn}(V_{li})$ and the rest to zero. Define $\widetilde{V} := \frac{1}{q}\sum_{(i,l) \in Q} X_{li}$. Then $\mathbb{E}[\widetilde{V}] = V$. Let also $\overline{Q} := \{i | (i,l) \in Q \text{ for some } l \in [d]\}$ for convenience.

The outputs of the standard $k$-means and Algorithm 2 can be stated, respectively, as $c_j^{*\,t+1} = V\chi_j^t$ and $(c_j^{t+1})_l = \lambda_j(\widetilde{V}\chi_j^t)_l = \frac{p}{n|P_j|}\sum_{(i,l') \in Q_{jl}}\frac{\|V\|_{1,1}}{q}\mathrm{sgn}(V_{l'i})$, where $Q_{jl} := \{(i,l') \in Q | (\ell_i^t, l') = (j,l)\}$. In order to bound $\|c_j^{*\,t+1} - c_j^{t+1}\|$, once again, by the triangle inequality,

$$\|c_j^{*\,t+1} - c_j^{t+1}\| \leq |\lambda_j - 1|\|V\chi_j^t\| + |\lambda_j|\|(\widetilde{V} - V)\chi_j^t\|,$$

and we just need to show that $|\lambda_j - 1| \leq \frac{\varepsilon}{2\|V\chi_j^t\|}$ and $\|(\widetilde{V} - V)\chi_j^t\| \leq \frac{\varepsilon}{2|\lambda_j|}$. Exactly as in Theorem 8, it suffices to take $p = O\big(\frac{\|V\|^2}{n}\frac{k^2}{\varepsilon^2}\log\frac{k}{\delta}\big)$ in order to bound $|\lambda_j - 1| \leq \frac{\varepsilon}{2\|V\chi_j^t\|}$ with probability $1 - \frac{\delta}{4}$.

Regarding the bound on $\widetilde{V}$, we use Freedman's inequality to prove $\|(\widetilde{V} - V)\chi_j^t\| \leq \frac{\varepsilon}{4} \leq \frac{\varepsilon}{2|\lambda_j|}$ (using that $|\lambda_j| \leq 2$). For such, let $f(X_1, \ldots, X_q) = \|(\widetilde{V} - V)\chi_j^t\|$. First, for all $i \in [q]$,

$$|f(X_1, \ldots, X_i, \ldots, X_q) - f(X_1, \ldots, X_i', \ldots, X_q)| = \big|\|(\widetilde{V} - V)\chi_j^t\| - \|(\widetilde{V}' - V)\chi_j^t\|\big|$$

$$\leq \|(\widetilde{V} - \widetilde{V}')\chi_j^t\| = \frac{1}{q}\|(X_i - X_i')\chi_j^t\| \leq \frac{2}{q}\|X_i\chi_j^t\| \leq \frac{2\|V\|_{1,1}}{q|\mathcal{C}_j^t|}.$$

Second, we bound the variance: for all $i \in [q]$,

$$\mathop{\mathbb{E}}_{X_i, X_i'}\big[f(X_1, \ldots, X_i, \ldots, X_q) - f(X_1, \ldots, X_i', \ldots, X_q)\big]^2 \leq \frac{1}{q^2}\mathop{\mathbb{E}}_{X_i, X_i'}\big[\|(X_i - X_i')\chi_j^t\|\big]^2$$

$$\leq \frac{4}{q^2}\mathop{\mathbb{E}}_{X_i}[\|X_i\chi_j^t\|^2] = \frac{4}{q^2}(\chi_j^t)^\top \mathop{\mathbb{E}}_{X}[X^\top X]\chi_j^t = \frac{4}{q^2}(\chi_j^t)^\top \|V\|_{1,1}\,\mathrm{diag}((\|v_i\|_1)_{i \in [n]})\chi_j^t \leq \frac{4\|V\|_{1,1}^2}{q^2|\mathcal{C}_j^t|^2}.$$

We employ Freedman's inequality with the Doob martingale $Y_i := \mathbb{E}[f(X_1, \ldots, X_q) | X_1, \ldots, X_i]$ for $i \in [q]$. Then Fact 6 with $B = \frac{2\|V\|_{1,1}}{q|\mathcal{C}_j^t|}$ and $\sigma^2 = q \cdot \frac{4\|V\|_{1,1}^2}{q^2|\mathcal{C}_j^t|^2}$ leads to

$$\Pr\bigg[\|(\widetilde{V} - V)\chi_j^t\| \geq \frac{\varepsilon}{4}\bigg] \leq \exp\left(-\frac{\varepsilon^2/32}{\frac{4\|V\|_{1,1}^2}{q|\mathcal{C}_j^t|^2} + \frac{\varepsilon\|V\|_{1,1}}{6q|\mathcal{C}_j^t|}}\right).$$

It suffices to take $q = O\big(\max\{\frac{\|V\|_{1,1}^2}{n^2}\frac{k^2}{\varepsilon^2}, \frac{\|V\|_{1,1}}{n}\frac{k}{\varepsilon}\}\log\frac{k}{\delta}\big)$ to approximate $\|(\widetilde{V} - V)\chi_j^t\| \leq \frac{\varepsilon}{2}$ with probability at least $1 - \frac{\delta}{4k}$ (already using that $|\mathcal{C}_j^t| = \Omega(\frac{n}{k})$). All in all, we have $\|c_j^{*\,t+1} - c_j^{t+1}\| \leq \varepsilon$ with probability at least $1 - \frac{\delta}{2}$ for all the centroids (using a union bound).

We now turn our attention to the query and time complexities. Another main difference to Theorem 8 is that the clusters $\{\mathcal{C}_j^t\}_{j \in [k]}$ are computed by approximating the distances $\|v_i - c_j^t\|$ using an $\ell_2$-sampling procedure explained in Lemma 11. More precisely, for any $i \in [n]$ we can compute

$$\ell_i^t \in \bigg\{j \in [k] : \|v_i - c_j^t\|^2 \leq \min_{j' \in [k]} \|v_i - c_{j'}^t\|^2 + \nu\bigg\}$$

with probability $1 - \frac{\delta}{4(p+q)}$ in $\widetilde{O}\big( \frac{\|V\|_F^2}{n} \frac{k \|v_i\|^2}{\nu^2} \log n \big)$ time and using $\widetilde{O}\big( \frac{\|V\|_F^2}{n} \frac{k \|v_i\|^2}{\nu^2} \big)$ queries, where $\widetilde{O}(\cdot)$ hides $\operatorname{poly} \log$ factors in $k$, $d$, $\frac{1}{\delta}$, $\frac{1}{\nu}$, $\frac{\|V\|_F}{\sqrt{n}}$. The classical query complexity in computing $\{\ell_i^t\}_{i \in P \cup \overline{Q}}$ is thus $\widetilde{O}\big( \frac{\|V\|_F^2}{n} \frac{k}{\nu^2} \sum_{i \in P \cup \overline{Q}} \|v_i\|^2 \big) = \widetilde{O}\big( (p+q) \frac{\|V\|_F^2 \|V\|_{2,\infty}^2}{n} \frac{k}{\nu^2} \big)$, while the time complexity has an extra $O(\log n)$ factor.[2] In summary, the time and query complexities are:

- Sampling $P \subseteq [n], Q \subseteq [n] \times [d]$ takes $O(p+q)$ queries and $O((p+q) \log(nd))$ time;

- Computing the labels $\{\ell_i^t\}_{i \in P \cup \overline{Q}}$ takes $\widetilde{O}\big( (p+q) \frac{\|V\|_F^2 \|V\|_{2,\infty}^2}{n} \frac{k}{\nu^2} \big)$ classical queries and $\widetilde{O}\big( (p+q) \frac{\|V\|_F^2 \|V\|_{2,\infty}^2}{n} \frac{k}{\nu^2} \log n \big)$ time;

- Querying the entries $\{V_{li}\}_{(i,l) \in Q}$ and computing all new centroids $\{c_j^{t+1}\}_{j \in [k]}$ takes $O(q)$ queries and $O(q \log(nd))$ time.

The total query complexity is thus $\widetilde{O}\big( (p+q) \frac{\|V\|_F^2 \|V\|_{2,\infty}^2}{n} \frac{k}{\nu^2} \big)$ and the time complexity is

$$\widetilde{O}\left( (p+q) \frac{\|V\|_F^2 \|V\|_{2,\infty}^2}{n} \frac{k}{\nu^2} \log n \right) = \widetilde{O}\left( \left( \frac{\|V\|^2}{n} + \frac{\|V\|_{1,1}^2}{n^2} \right) \frac{\|V\|_F^2 \|V\|_{2,\infty}^2}{n} \frac{k^3}{\varepsilon^2 \nu^2} \log n \right). \quad \square$$

**Lemma 11** (Approximate classical cluster assignment). *Assume classical query access to matrix $V = [v_1, \ldots, v_n] \in \mathbb{R}^{d \times n}$. Let $\delta \in (0,1)$, $\nu > 0$, and $0 < \varepsilon \leq \frac{\|V\|_F}{\sqrt{n}}$. Consider the centroid matrix $C^t = [c_1^t, \ldots, c_k^t] \in \mathbb{R}^{d \times k}$ such that $\big\| c_j^t - |\mathcal{C}_j^{t-1}|^{-1} \sum_{i \in \mathcal{C}_j^{t-1}} v_i \big\| \leq \varepsilon$ with $|\mathcal{C}_j^{t-1}| = \Omega(\frac{n}{k})$ for all $j \in [k]$. For any $i \in [n]$, there is a classical algorithm that outputs $\ell_i^t \in \{j \in [k] : \|v_i - c_j^t\|^2 \leq \min_{j' \in [k]} \|v_i - c_{j'}^t\|^2 + \nu\}$ with probability $1 - \delta$ in $O\big( \frac{\|V\|_F^2}{n} \frac{k \|v_i\|^2}{\nu^2} \log \frac{k}{\delta} \log(nd) \big)$ time and using $O\big( \frac{\|V\|_F^2}{n} \frac{k \|v_i\|^2}{\nu^2} \log \frac{k}{\delta} \big)$ classical queries.*

*Proof.* Fix $(i,j) \in [n] \times [k]$ and consider the random variable $X^{(ij)}$ such that

$$\Pr\left[ X^{(ij)} = \|v_i\|^2 \frac{(c_j^t)_l}{(v_i)_l} \right] = \frac{(v_i)_l^2}{\|v_i\|^2} \qquad \forall l \in [d].$$

We can straightforwardly calculate

$$\mathbb{E}[X^{(ij)}] = \sum_{l \in [d]} \|v_i\|^2 \frac{(c_j^t)_l}{(v_i)_l} \frac{(v_i)_l^2}{\|v_i\|^2} = \sum_{l \in [d]} (c_j^t)_l (v_i)_l = \langle c_j^t, v_i \rangle,$$

$$\operatorname{Var}[X^{(ij)}] \leq \sum_{l \in [d]} \|v_i\|^4 \frac{(c_j^t)_l^2}{(v_i)_l^2} \frac{(v_i)_l^2}{\|v_i\|^2} = \|v_i\|^2 \sum_{l \in [d]} (c_j^t)_l^2 = \|v_i\|^2 \|c_j^t\|^2.$$

By taking the median of $K = 8 \ln \frac{k}{\delta}$ copies of the mean of $\frac{64}{\nu^2} \|v_i\|^2 \|c_j^t\|^2$ copies of $X^{(ij)}$, we obtain an estimate of $\langle c_j^t, v_i \rangle$ within additive error $\frac{\nu}{4}$ with probability $1 - \frac{\delta}{k}$ (Fact 5). From this, we can output $w_{ij} \in \mathbb{R}$ such that $|w_{ij} - \|v_i - c_j^t\|^2| \leq \frac{\nu}{2}$ with probability $1 - \frac{\delta}{k}$. Let $\ell_i^t = \arg \min_{j \in [k]} w_{ij}$. Then $\ell_i^t \in \{j \in [k] : \|v_i - c_j^t\|^2 \leq \min_{j' \in [k]} \|v_i - c_{j'}^t\|^2 + \nu\}$ with probability $1 - \delta$ by a union bound.

---

[2] It is possible to do slightly better than $\sum_{i \in P \cup \overline{Q}} \|v_i\|^2 \leq (p+q) \|V\|_{2,\infty}^2$ via concentration bounds. Let the random variable $\Pr\big[ Y(i) = \frac{\|v_i\|^2}{\|V\|_{2,\infty}^2} \big] = \frac{1}{n}$ with mean $\mathbb{E}[Y] = \frac{\|V\|_F^2}{n \|V\|_{2,\infty}^2}$. By a Chernoff's bound, $\Pr\big[ \sum_{i \in P} \|v_i\|^2 \geq 2p \frac{\|V\|_F^2}{n} \big] \leq \exp\big( -\frac{p}{3} \frac{\|V\|_F^2}{n \|V\|_{2,\infty}^2} \big) \leq \frac{\delta}{2}$ if $p \geq 3 \frac{n \|V\|_{2,\infty}^2}{\|V\|_F^2} \ln \frac{2}{\delta}$. On the other hand, let the random variable $\Pr\big[ Y(i) = \frac{\|v_i\|^2}{\|V\|_{2,\infty}^2} \big] = \frac{\|v_i\|}{\|V\|_{2,1}}$ with mean $\mathbb{E}[Y] = \frac{1}{\|V\|_{2,1}} \sum_{i \in [n]} \frac{\|v_i\|^3}{\|V\|_{2,\infty}^2} \leq \frac{\|V\|_F^2}{\|V\|_{2,1} \|V\|_{2,\infty}^2}$. By a Chernoff's bound, $\Pr\big[ \sum_{i \in \overline{Q}} \|v_i\|^2 \geq 2q \frac{\|V\|_F^2 \|V\|_{2,\infty}}{\|V\|_{2,1}} \big] \leq \exp\big( -\frac{q}{3} \frac{\|V\|_F^2}{\|V\|_{2,1} \|V\|_{2,\infty}} \big) \leq \frac{\delta}{2}$ if $q \geq 3 \frac{\|V\|_{2,1} \|V\|_{2,\infty}}{\|V\|_F^2} \ln \frac{2}{\delta}$. Thus $\sum_{i \in P \cup \overline{Q}} \|v_i\|^2 \leq 2 \big( p + q \frac{n \|V\|_{2,\infty}}{\|V\|_{2,1}} \big) \frac{\|V\|_F^2}{n}$ with probability $1 - \delta$ for large enough $p$ and $q$.

Regarding the sample and time complexities, the total amount of samples is

$$O\left(\frac{\|v_i\|^2}{\nu^2}\log\frac{k}{\delta}\sum_{j\in[k]}\|c_j^t\|^2\right) = O\left(\frac{\|v_i\|^2\|C^t\|_F^2}{\nu^2}\log\frac{k}{\delta}\right) = O\left(\frac{\|v_i\|^2}{\nu^2}\frac{k\|V\|_F^2}{n}\log\frac{k}{\delta}\right),$$

where we used that $\|c_j^t\| \le \varepsilon + |\mathcal{C}_j^{t-1}|^{-1}\sum_{i\in\mathcal{C}_j^{t-1}}\|v_i\|$ implies

$$\|C^t\|_F^2 \le 2k\varepsilon^2 + \sum_{j\in[k]}\frac{2}{|\mathcal{C}_j^{t-1}|^2}\left(\sum_{i\in\mathcal{C}_j^{t-1}}\|v_i\|\right)^2 \le 2k\varepsilon^2 + \sum_{j\in[k]}\frac{2}{|\mathcal{C}_j^{t-1}|}\sum_{i\in\mathcal{C}_j^{t-1}}\|v_i\|^2 = O\left(\frac{k\|V\|_F^2}{n}\right),$$

using that $|\mathcal{C}_j^{t-1}| = \Omega(\frac{n}{k})$ for all $j \in [k]$ and $\varepsilon \le \frac{\|V\|_F}{\sqrt{n}}$. The total time complexity is simply the sample complexity times $O(\log(nd))$. $\qquad\square$

# B QUANTUM ALGORITHMS

We now describe our quantum $(\varepsilon, \nu)$-$k$-means algorithms. Similarly to our classical algorithm from the previous section, we approximate the quantities $\sum_{i\in\mathcal{C}_j^t}v_i$ and $|\mathcal{C}_j^t|$ for all $j \in [k]$ separately. This time, however, we employ quantum query access from Definition 3 to build quantum unitaries which are fed into the multivariate quantum mean estimator from Cornelissen et al. (2022). As an intermediary step, the quantities $\ell_i^t = \arg\min_{j\in[k]}\|v_i - c_j^t\|$ are computed in superposition as part of these unitaries (Lemma 16).

Before presenting and proving the correctness of our quantum algorithm, recall a few subroutines that will serve as building blocks: the quantum minimum finding subroutine from Dürr & Høyer (1996), its generalisation with variable times due to Ambainis (2010; 2012), and the multivariate quantum mean estimation subroutine from Cornelissen et al. (2022).

**Fact 12** (Quantum min-finding (Dürr & Høyer, 1996)). *Given $\delta \in (0,1)$ and oracle $U_x : |i\rangle|\bar{0}\rangle \mapsto |i\rangle|x_i\rangle$ for $x \in \mathbb{R}^N$, there is a quantum algorithm that outputs $|\Psi_x\rangle$ using $O(\sqrt{N}\log\frac{1}{\delta})$ queries to $U_x$ such that, upon measuring $|\Psi_x\rangle$ in the computational basis, the outcome is $\arg\min_{i\in[N]}x_i$ with probability $1-\delta$.*

**Fact 13** (Variable-time quantum min-finding (Ambainis, 2010)). *Let $\delta \in (0,1)$, $x \in \mathbb{R}^N$, and $\{U_i\}_{i\in[N]}$ a collection of oracles such that $U_i : |\bar{0}\rangle \mapsto |x_i\rangle$ in time $O(t_i)$. There is a quantum algorithm that runs in time $O((\sum_{i\in[N]}t_i^2)^{\frac{1}{2}}\log\frac{1}{\delta})$ and outputs $|\Psi_x\rangle$ such that, upon measuring $|\Psi_x\rangle$ in the computational basis, the outcome is $\arg\min_{i\in[N]}x_i$ with probability $1-\delta$.*

**Fact 14** ((Cornelissen et al., 2022, Theorem 3.3)). *Consider a bounded $N$-dimensional random variable $X : \Omega \to \mathbb{R}^N$ over a probability space $(\Omega, 2^\Omega, \mathbb{P})$ with mean $\mu = \sum_{\omega\in\Omega}\mathbb{P}(\omega)X(\omega)$ and such that $\|X\| \le 1$. Assume access to unitaries $U_\mathbb{P} : |\bar{0}\rangle \mapsto \sum_{\omega\in\Omega}\sqrt{\mathbb{P}(\omega)}|\omega\rangle$ and $\mathcal{B}_X : |\omega\rangle|\bar{0}\rangle \mapsto |\omega\rangle|X(\omega)\rangle$. Given $\delta \in (0,1)$, $m \in \mathbb{N}$, and an upper bound $L_2 \ge \mathbb{E}[\|X\|]$, there is a quantum algorithm that outputs $\widetilde{\mu} \in \mathbb{R}^N$ such that $\|\mu - \widetilde{\mu}\|_\infty \le \frac{\sqrt{L_2}\log(N/\delta)}{m}$ with success probability at least $1-\delta$, using $O(m\operatorname{poly}\log m)$ queries to the oracles $U_\mathbb{P}$ and $\mathcal{B}_X$, and in time $\widetilde{O}(mN)$.[3]*

**Theorem 15** (Quantum $(\varepsilon, 0)$-$k$-means algorithm). *Let $\varepsilon > 0$, $\delta \in (0,1)$, and assume quantum query access to $V = [v_1, \ldots, v_n] \in \mathbb{R}^{d\times n}$. If all clusters satisfy $|\mathcal{C}_j^t| = \Omega(\frac{n}{k})$, then Algorithm 3 outputs centroids consistent with the $(\varepsilon, \nu = 0)$-$k$-means with probability $1-\delta$. The complexities*

---

[3]The time complexity of the mean estimation subroutine is not analysed in Cornelissen et al. (2022), so we give a sketch of its analysis here. The last step of the algorithm needs to perform $N$ parallel inverse QFTs on $m$ qubits, which requires $\widetilde{\Theta}(mN)$ gates since we need to touch every qubit at least once. It remains to show that we can implement the rest of the algorithm in time $\widetilde{O}(mN)$. In the preprocessing step, we compute the $\ell_2$-norm of the random variable in time $O(N)$, a total of $\widetilde{O}(m)$ times. The main routine, subsequently, runs with $m$ repetitions, within each of which we perform arithmetic operations such as computing the inner product of two $N$-dimensional vectors, in $O(N)$ time, and do quantum singular value transformations. This last step is used to turn a so-called probability oracle into a phase oracle, and takes $\widetilde{O}(1)$ time to implement. The total time complexity of this step thus also becomes $\widetilde{O}(mN)$.

---

**Algorithm 3** Quantum $(\varepsilon, \nu = 0)$-$k$-means algorithm

---

**Input:** Quantum query access to data matrix $V = [v_1, \ldots, v_n] \in \mathbb{R}^{d \times n}$, parameters $\delta, \varepsilon$.
1: Select $k$ initial centroids $c_1^0, \ldots, c_k^0$
2: **for** $t = 0$ until convergence **do**
3:     Build quantum query access to $[c_1^t, \ldots, c_k^t] \in \mathbb{R}^{d \times k}$
4:     Using Lemma 16 to obtain $|i\rangle|\bar{0}\rangle \mapsto |i\rangle|\ell_i^t\rangle$ where $\ell_i^t = \arg\min_{j \in [k]} \|v_i - c_j^t\|$, construct the unitaries

$$U_I : |\bar{0}\rangle \mapsto \sum_{i \in [n]} \frac{1}{\sqrt{n}} |i, \ell_i^t\rangle, \qquad U_V : |\bar{0}\rangle \mapsto \sum_{i \in [n]} \sqrt{\frac{\|v_i\|}{\|V\|_{2,1}}} |i, \ell_i^t\rangle,$$

$$\mathcal{B}_I : |i, j\rangle|\bar{0}\rangle^{\otimes k} \mapsto |i, j\rangle|\bar{0}\rangle^{\otimes(j-1)}|1\rangle|\bar{0}\rangle^{\otimes(k-j-1)},$$

$$\mathcal{B}_V : |i, j\rangle|\bar{0}\rangle^{\otimes kd} \mapsto |i, j\rangle|\bar{0}\rangle^{\otimes(j-1)d}|v_i/\|v_i\|\rangle|\bar{0}\rangle^{\otimes(k-j-1)d}$$

5:     Apply the multivariate quantum mean estimator (Fact 14) with $p = \widetilde{O}\big(\frac{\|V\|}{\sqrt{n}} \frac{k^{3/2}}{\varepsilon}\big)$ queries to the unitaries $U_I$ and $\mathcal{B}_I$ to obtain $P \in \mathbb{R}^k$
6:     Apply the multivariate quantum mean estimator (Fact 14) with $q = \widetilde{O}\big(\frac{\|V\|_{2,1}}{\sqrt{n}} \frac{k\sqrt{d}}{\varepsilon}\big)$ queries to the unitaries $U_V$ and $\mathcal{B}_V$ to obtain $Q \in (\mathbb{R}^d)^k$
7:     For $j \in [k]$, record the new centroids $c_j^{t+1} = \frac{\|V\|_{2,1}}{n} \frac{Q_j}{P_j}$
8: **end for**

---

per iteration of Algorithm 3 are (up to polylog factors in $k$, $d$, $\frac{1}{\delta}$, $\frac{1}{\varepsilon}$, $\frac{\|V\|_F}{\sqrt{n}}$)

$$\textbf{\textit{Quantum queries:}} \ \widetilde{O}\bigg(\bigg(\sqrt{k}\frac{\|V\|}{\sqrt{n}} + \sqrt{d}\frac{\|V\|_{2,1}}{n}\bigg)\frac{k^{\frac{3}{2}}d}{\varepsilon}\bigg),$$

$$\textbf{\textit{Time:}} \ \widetilde{O}\bigg(\bigg(\sqrt{k}\frac{\|V\|}{\sqrt{n}} + \sqrt{d}\frac{\|V\|_{2,1}}{n}\bigg)\frac{k^{\frac{3}{2}}d}{\varepsilon}(\sqrt{k} + \log n)\bigg).$$

*Proof.* We start with the error analysis. Consider the unitaries

$$U_V : |\bar{0}\rangle \mapsto \sum_{i \in [n]} \sqrt{\frac{\|v_i\|}{\|V\|_{2,1}}} |i, \ell_i^t\rangle, \qquad \mathcal{B}_V : |i, j\rangle|\bar{0}\rangle^{\otimes kd} \mapsto |i, j\rangle|\bar{0}\rangle^{\otimes(j-1)d}|v_i/\|v_i\|\rangle|\bar{0}\rangle^{\otimes(k-j-1)d},$$

$$U_I : |\bar{0}\rangle \mapsto \sum_{i \in [n]} \frac{1}{\sqrt{n}} |i, \ell_i^t\rangle, \qquad \mathcal{B}_I : |i, j\rangle|\bar{0}\rangle^{\otimes k} \mapsto |i, j\rangle|\bar{0}\rangle^{\otimes(j-1)}|1\rangle|\bar{0}\rangle^{\otimes(k-j-1)}.$$

The unitaries $U_V$ and $U_I$ can be thought of as preparing a superposition over probability spaces with distributions $\mathbb{P}_V$ and $\mathbb{P}_I$, respectively, given by

$$\mathbb{P}_V(i,j) = \begin{cases} \frac{\|v_i\|}{\|V\|_{2,1}} & \text{if } j = \ell_i^t, \\ 0 & \text{if } j \neq \ell_i^t, \end{cases} \qquad \text{and} \qquad \mathbb{P}_I(i,j) = \begin{cases} \frac{1}{n} & \text{if } j = \ell_i^t, \\ 0 & \text{if } j \neq \ell_i^t, \end{cases}$$

while the unitaries $\mathcal{B}_V$ and $\mathcal{B}_I$ can be thought of as binary encoding the random variables $X_V : [n] \times [k] \to (\mathbb{R}^d)^k$ and $X_I : [n] \times [k] \to \mathbb{R}^k$, respectively, given by $X_V(i,j) = (0, \ldots, 0, \frac{v_i}{\|v_i\|}, 0, \ldots, 0)$ and $X_I(i,j) = (0, \ldots, 0, 1, 0, \ldots, 0)$, where the non-zero entry is the $j$-th entry. Note that

$$\sum_{(i,j) \in [n] \times [k]} \mathbb{P}_V(i,j) X_V(i,j) = \bigg(\sum_{i \in \mathcal{C}_1^t} \frac{v_i}{\|V\|_{2,1}}, \ldots, \sum_{i \in \mathcal{C}_k^t} \frac{v_i}{\|V\|_{2,1}}\bigg),$$

$$\sum_{(i,j) \in [n] \times [k]} \mathbb{P}_I(i,j) X_I(i,j) = \bigg(\frac{|\mathcal{C}_1^t|}{n}, \ldots, \frac{|\mathcal{C}_k^t|}{n}\bigg).$$

Therefore, the multivariate quantum mean estimator (Fact 14) returns $P \in \mathbb{R}^k$ and $Q \in (\mathbb{R}^d)^k$ such that, with probability at least $1 - \delta$ and for some $\varepsilon_1, \varepsilon_2 > 0$ to be determined,

$$\left| P_j - \frac{|\mathcal{C}_j^t|}{n} \right| \leq \varepsilon_1 \qquad \text{and} \qquad \left\| Q_j - \sum_{i \in \mathcal{C}_j^t} \frac{v_i}{\|V\|_{2,1}} \right\|_\infty \leq \varepsilon_2 \qquad \forall j \in [k].$$

This means that, by a triangle inequality,

$$\|c_j^{*\,t+1} - c_j^{t+1}\| \leq \left| \frac{|\mathcal{C}_j^t|}{nP_j} - 1 \right| \left\| \frac{1}{|\mathcal{C}_j^t|} \sum_{i \in \mathcal{C}_j^t} v_i \right\| + \frac{\|V\|_{2,1}}{nP_j} \left\| Q_j - \sum_{i \in \mathcal{C}_j^t} \frac{v_i}{\|V\|_{2,1}} \right\|.$$

For $\varepsilon_1$ small enough such that $\varepsilon_1 \leq \min_{j \in [k]} \frac{|\mathcal{C}_j^t|}{2n}$, then $\left| P_j - \frac{|\mathcal{C}_j^t|}{n} \right| \leq \varepsilon_1 \implies \frac{1}{nP_j} \leq \frac{1}{|\mathcal{C}_j^t| - n\varepsilon_1} \leq \frac{2}{|\mathcal{C}_j^t|}$ and $\left| \frac{1}{P_j} - \frac{n}{|\mathcal{C}_j^t|} \right| \leq \frac{2n^2}{|\mathcal{C}_j^t|^2} \varepsilon_1$ according to Claim 7. Moreover, $\left\| \frac{1}{|\mathcal{C}_j^t|} \sum_{i \in \mathcal{C}_j^t} v_i \right\| = \|V\chi_j^t\| \leq \|V\| \|\chi_j^t\| = \|V\|/\sqrt{|\mathcal{C}_j^t|}$. Hence

$$\|c_j^{*\,t+1} - c_j^{t+1}\| \leq \frac{\|V\|}{\sqrt{|\mathcal{C}_j^t|}} \frac{2n}{|\mathcal{C}_j^t|} \varepsilon_1 + \frac{2\sqrt{d}\|V\|_{2,1}}{|\mathcal{C}_j^t|} \varepsilon_2.$$

It suffices to take $\varepsilon_1 = O\left( \frac{\sqrt{n}}{\|V\|} \frac{\varepsilon}{k^{3/2}} \right)$ and $\varepsilon_2 = O\left( \frac{n}{\|V\|_{2,1}} \frac{\varepsilon}{k\sqrt{d}} \right)$ in order to obtain $\|c_j^{*\,t+1} - c_j^{t+1}\| \leq \varepsilon$, where we used that $|\mathcal{C}_j^t| = \Omega(\frac{n}{k})$. In order to obtain $\varepsilon_1 = O\left( \frac{\sqrt{n}}{\|V\|} \frac{\varepsilon}{k^{3/2}} \right)$, we must query the unitaries $U_I$ and $\mathcal{B}_I$ in the multivariate quantum mean estimator $p = \widetilde{O}\left( \frac{\|V\|}{\sqrt{n}} \frac{k^{3/2}}{\varepsilon} \right)$ times (since $\|X_I\| = 1$ and $\mathbb{E}[\|X_I\|] = 1$). On the other hand, in order to obtain $\varepsilon_2 = O\left( \frac{n}{\|V\|_{2,1}} \frac{\varepsilon}{k\sqrt{d}} \right)$, we must query the unitaries $U_V$ and $\mathcal{B}_V$ in the multivariate quantum mean estimator $q = \widetilde{O}\left( \frac{\|V\|_{2,1}}{n} \frac{k\sqrt{d}}{\varepsilon} \right)$ times (since $\|X_V\| = 1$ and $\mathbb{E}[\|X_V\|] = 1$).

Finally, we must show how to perform the unitaries $U_V, U_I, \mathcal{B}_V, \mathcal{B}_I$. The binary-encoding unitary $\mathcal{B}_V$ is $d$ QRAM calls ($O(d \log n)$ time), followed by a normalisation computation ($O(d)$ time), followed by $d$ controlled-SWAPs on $k$ qubits ($O(kd \log k)$ time (Berry et al., 2015)), while $\mathcal{B}_I$ is simply 1 controlled-SWAP on $k$ qubits. On the other hand, the probability-distribution-encoding unitaries $U_V, U_I$ can be performed via the initial state preparations $|\bar{0}\rangle \mapsto \sum_{i \in [n]} \sqrt{\frac{\|v_i\|}{\|V\|_{2,1}}} |i\rangle$ and $|\bar{0}\rangle \mapsto \frac{1}{\sqrt{n}} \sum_{i \in [n]} |i\rangle$, respectively, followed by the mapping $|i\rangle|\bar{0}\rangle \mapsto |i\rangle|\ell_i^t\rangle$. In Lemma 16 we show how to implement the mapping $|i\rangle|\bar{0}\rangle \mapsto |i\rangle|\ell_i^t\rangle$ in $O(\sqrt{k}d \log \frac{1}{\delta} \log n)$ time using $O(\sqrt{k}d \log \frac{1}{\delta})$ quantum queries. In summary,

1. $\mathcal{B}_V$ requires $O(d)$ quantum queries and $O(d \log n + kd \log k)$ time;[4]

2. $U_V$ requires $O(\sqrt{k}d \log \frac{1}{\delta})$ quantum queries and $O(\sqrt{k}d \log \frac{1}{\delta} \log n)$ time;

3. $\mathcal{B}_I$ requires no quantum queries and $O(k \log k)$ time;

4. $U_I$ requires $O(\sqrt{k}d \log \frac{1}{\delta})$ quantum queries and $O(\sqrt{k}d \log \frac{1}{\delta} \log n)$ time.

Collecting all the terms, the total number of quantum queries is $\widetilde{O}((p+q)\sqrt{k}d)$, while the overall time complexity is

$$\widetilde{O}\left( (p+q)\sqrt{k}d(\sqrt{k} + \log n) \right) = \widetilde{O}\left( \left( \sqrt{k}\frac{\|V\|}{\sqrt{n}} + \sqrt{d}\frac{\|V\|_{2,1}}{n} \right) \frac{k^{\frac{3}{2}}d}{\varepsilon}(\sqrt{k} + \log n) \right). \qquad \square$$

---

[4]If one has access to a quantum random access gate (Ambainis, 2007; Allcock et al., 2024), which is the unitary that performs $|i\rangle|b\rangle|x_1, \ldots, x_N\rangle \mapsto |i\rangle|x_i\rangle|x_1, \ldots, x_{i-1}, b, x_{i+1}, \ldots, x_N\rangle$ in $O(\log N)$ time, then $\mathcal{B}_V$ requires $O(d \log n)$ time and the final runtime of Algorithm 3 becomes $\widetilde{O}\left( \left( \sqrt{k}\frac{\|V\|}{\sqrt{n}} + \sqrt{d}\frac{\|V\|_{2,1}}{n} \right) \frac{k^{3/2}d}{\varepsilon} \log n \right)$.

**Lemma 16** (Exact quantum cluster assignment). *Let $\delta \in (0,1)$ and assume quantum query access to matrices $V = [v_1, \ldots, v_n] \in \mathbb{R}^{d \times n}$ and $[c_1^t, \ldots, c_k^t] \in \mathbb{R}^{d \times k}$. There is a quantum algorithm that performs the mapping $|i\rangle|\bar{0}\rangle \mapsto |i\rangle|L_i^t\rangle$ using $O(\sqrt{k}d\log\frac{1}{\delta})$ quantum queries and in $O(\sqrt{k}d\log\frac{1}{\delta}\log n)$ time such that, upon measuring $|L_i^t\rangle$ on the computational basis, the outcome equals $\arg\min_{j\in[k]} \|v_i - c_j^t\|$ with probability at least $1 - \delta$.*

*Proof.* First we describe how to perform the mapping $|i,j\rangle|\bar{0}\rangle \mapsto |i,j\rangle|\|v_i - c_j^t\|\rangle$. Starting from $|i,j\rangle|\bar{0},\bar{0}\rangle|\bar{0}\rangle$, we query $2d$ times the QRAM oracles used in Definition 3 to map

$$|i,j\rangle|\bar{0},\bar{0}\rangle|\bar{0}\rangle \mapsto |i,j\rangle|v_i, c_j^t\rangle|\bar{0}\rangle.$$

This operation is followed by computing the distance $\|v_i - c_j^t\|$ between the vectors $v_i$ and $c_j^t$ in $O(d)$ size and $O(\log d)$ depth by using a classical circuit, which leads to

$$|i,j\rangle|v_i, c_j^t\rangle|\bar{0}\rangle \mapsto |i,j\rangle|v_i, c_j^t\rangle|\|v_i - c_j^t\|\rangle.$$

Uncomputing the first step leads to the desire state. Overall, the map $|i,j\rangle|\bar{0}\rangle \mapsto |i,j\rangle|\|v_i - c_j^t\|\rangle$ uses $O(d)$ queries to the matrices $V$ and $[c_1^t, \ldots, c_k^t]$.

Fix $i \in [n]$. The mapping $|j\rangle|\bar{0}\rangle \mapsto |j\rangle|\|v_i - c_j^t\|\rangle$ can be viewed as quantum access to the vector $(\|v_i - c_j^t\|)_{j\in[k]}$. Therefore, we can assign a cluster $\ell_i^t := \arg\min_{j\in[k]} \|v_i - c_j^t\|$ to the vector $v_i$ by using (controlled on $|i\rangle$) quantum minimum finding subroutine (Fact 12), which leads to the map $|i\rangle|\bar{0}\rangle \mapsto |i\rangle|L_i^t\rangle$, where upon measuring $|L_i^t\rangle$ on the computational basis, the outcome equals $\ell_i^t = \arg\min_{j\in[k]} \|v_i - c_j^t\|$ with probability at least $1 - \delta$. The time cost of finding the minimum is $O(\sqrt{k}\log\frac{1}{\delta})$ queries to the unitary performing the mapping $|i,j\rangle|\bar{0}\rangle \mapsto |i,j\rangle|\|v_i - c_j^t\|\rangle$, to a total time complexity of $O(\sqrt{k}d\log\frac{1}{\delta}\log n)$ and $O(\sqrt{k}d\log\frac{1}{\delta})$ quantum queries. $\square$

**Remark 1.** *It is possible to avoid QRAM access to the centroids $[c_1^t, \ldots, c_k^t] \in \mathbb{R}^{d \times k}$ by accessing them through the fixed registers $\bigotimes_{j\in[k]} |c_j^t\rangle$. This, however, hinders the use of quantum minimum finding. The index $\ell_i^t = \arg\min_{j\in[k]} \|v_i - c_j^t\|$ can be found, instead, through a classical circuit on the registers $\bigotimes_{j\in[k]} |\|v_i - c_j^t\|\rangle$, which modifies the time complexity of Lemma 16 to $O(d(k+\log n))$.*

Similarly to classical $(\varepsilon, \nu)$-$k$-means algorithm, it is possible to approximate the distances $\|v_i - c_j^t\|$ within quantum minimum finding using inherently quantum subroutines (Lemma 18) instead of a classical arithmetic circuit as in Algorithm 3. This allows us to replace the $O(d)$ time overhead with some norm of $V$. Algorithm 4 describes how this can be performed and the next theorem analyses its query and time complexities.

**Theorem 17** (Quantum $(\varepsilon, \nu)$-$k$-means algorithm). *Let $\varepsilon, \nu > 0$, $\delta \in (0,1)$, and assume quantum query access to $V = [v_1, \ldots, v_n] \in \mathbb{R}^{d \times n}$. If all clusters satisfy $|\mathcal{C}_j^t| = \Omega(\frac{n}{k})$, then Algorithm 4 outputs centroids consistent with the $(\varepsilon, \nu)$-$k$-means with probability $1 - \delta$. The complexities per iteration of Algorithm 4 are (up to polylog factors in $k$, $d$, $\frac{1}{\delta}$, $\frac{1}{\nu}$, $\frac{1}{\varepsilon}$, $\frac{\|V\|_F}{\sqrt{n}}$)*

$$\textbf{Quantum queries: } \widetilde{O}\left(\left(\sqrt{k}\frac{\|V\|}{\sqrt{n}} + \sqrt{d}\frac{\|V\|_{1,1}}{n}\right)\frac{\|V\|_F\|V\|_{2,\infty}}{\sqrt{n}}\frac{k^{\frac{3}{2}}}{\varepsilon\nu}\right),$$

$$\textbf{Time: } \widetilde{O}\left(\left(\sqrt{k}\frac{\|V\|}{\sqrt{n}} + \sqrt{d}\frac{\|V\|_{1,1}}{n}\right)\left(\frac{\|V\|_F\|V\|_{2,\infty}}{\sqrt{n}}\frac{k^{\frac{3}{2}}}{\varepsilon\nu}\log n + \frac{k^2 d}{\varepsilon}\right)\right).$$

*Proof.* The proof is similar to Theorem 15. The unitaries $U_I$ and $\mathcal{B}_I$ are still the same,

$$U_I : |\bar{0}\rangle \mapsto \sum_{i\in[n]} \frac{1}{\sqrt{n}}|i, \ell_i^t\rangle, \qquad \mathcal{B}_I : |i,j\rangle|0\rangle^{\otimes k} \mapsto |i,j\rangle|0\rangle^{\otimes(j-1)}|1\rangle|0\rangle^{\otimes(k-j-1)},$$

but the unitaries $U_V$ and $\mathcal{B}_V$ are now replaced with

$$U_V : |\bar{0}\rangle \mapsto \sum_{(i,l)\in[n]\times[d]} \sqrt{\frac{|V_{li}|}{\|V\|_{1,1}}}|i, \ell_i^t, l\rangle,$$

$$\mathcal{B}_V : |i,j,l\rangle|0\rangle^{\otimes kd} \mapsto |i,j,l\rangle|0\rangle^{\otimes((j-1)d+(l-1))}|\operatorname{sgn}(V_{li})\rangle|0\rangle^{\otimes((k-j-1)d+(d-l-1))}.$$

---

**Algorithm 4** Quantum $(\varepsilon, \nu)$-$k$-means algorithm

---

**Input:** Quantum query access to data matrix $V = [v_1, \ldots, v_n] \in \mathbb{R}^{d \times n}$, parameters $\delta, \varepsilon, \nu$.

1: Select $k$ initial centroids $c_1^0, \ldots, c_k^0$
2: **for** $t = 0$ until convergence **do**
3:      Build quantum query access to $[c_1^t, \ldots, c_k^t] \in \mathbb{R}^{d \times k}$
4:      Using Lemma 18 to obtain $|i\rangle|\bar{0}\rangle \mapsto |i\rangle|\ell_i^t\rangle$ such that $\ell_i^t \in \{j \in [k] : \|v_i - c_j^t\|^2 \leq \min_{j' \in [k]} \|v_i - c_{j'}^t\|^2 + \nu\}$, construct the unitaries

$$U_I : |\bar{0}\rangle \mapsto \sum_{i \in [n]} \frac{1}{\sqrt{n}} |i, \ell_i^t\rangle, \qquad U_V : |\bar{0}\rangle \mapsto \sum_{(i,l) \in [n] \times [d]} \sqrt{\frac{|V_{li}|}{\|V\|_{1,1}}} |i, \ell_i^t, l\rangle,$$

$$\mathcal{B}_I : |i,j\rangle|\bar{0}\rangle^{\otimes k} \mapsto |i,j\rangle|\bar{0}\rangle^{\otimes(j-1)}|1\rangle|\bar{0}\rangle^{\otimes(k-j)},$$

$$\mathcal{B}_V : |i,j,l\rangle|0\rangle^{\otimes kd} \mapsto |i,j,l\rangle|0\rangle^{\otimes((j-1)d+(l-1))}|\operatorname{sgn}(V_{li})\rangle|0\rangle^{\otimes((k-j-1)d+(d-l-1))}$$

5:      Apply the multivariate quantum mean estimator (Fact 14) with $p = \widetilde{O}\left(\frac{\|V\|}{\sqrt{n}} \frac{k^{3/2}}{\varepsilon}\right)$ queries to the unitaries $U_I$ and $\mathcal{B}_I$ to obtain $P \in \mathbb{R}^k$
6:      Apply the multivariate quantum mean estimator (Fact 14) with $q = \widetilde{O}\left(\frac{\|V\|_{1,1}}{\sqrt{n}} \frac{k\sqrt{d}}{\varepsilon}\right)$ queries to the unitaries $U_V$ and $\mathcal{B}_V$ to obtain $Q \in (\mathbb{R}^d)^k$
7:      For $j \in [k]$, record the new centroids $c_j^{t+1} = \frac{\|V\|_{1,1}}{n} \frac{Q_j}{P_j}$
8: **end for**

---

The new unitary $U_V$ can be thought of as preparing a superposition over the probability space with distribution $\mathbb{P}_V$ given by

$$\mathbb{P}_V(i, j, l) = \begin{cases} \frac{|V_{li}|}{\|V\|_{1,1}} & \text{if } j = \ell_i^t, \\ 0 & \text{if } j \neq \ell_i^t, \end{cases}$$

while the unitary $\mathcal{B}_V$ can be thought of as binary encoding the random variables $X_V : [n] \times [k] \times [d] \to (\mathbb{R}^d)^k$ given by $X_V(i, j, l) = (0, \ldots, 0, \operatorname{sgn}(V_{li}), 0, \ldots, 0)$, where the non-zero entry is the $((j-1)d + l)$-th entry. Note that

$$\sum_{(i,j,l) \in [n] \times [k] \times [d]} \mathbb{P}_V(i,j,l) X_V(i,j,l) = \left( \sum_{i \in \mathcal{C}_1^t} \frac{v_i}{\|V\|_{1,1}}, \ldots, \sum_{i \in \mathcal{C}_k^t} \frac{v_i}{\|V\|_{1,1}} \right).$$

Therefore, the multivariate quantum mean estimator (Fact 14) returns $P \in \mathbb{R}^k$ and $Q \in (\mathbb{R}^d)^k$ such that, with probability at least $1 - \delta$ and for some $\varepsilon_1, \varepsilon_2 > 0$ to be determined,

$$\left| P_j - \frac{|\mathcal{C}_j^t|}{n} \right| \leq \varepsilon_1 \qquad \text{and} \qquad \left\| Q_j - \sum_{i \in \mathcal{C}_j^t} \frac{v_i}{\|V\|_{1,1}} \right\|_\infty \leq \varepsilon_2 \qquad \forall j \in [k].$$

Similar to Theorem 15, by a triangle inequality,

$$\|c_j^{*t+1} - c_j^{t+1}\| \leq \frac{\|V\|}{\sqrt{|\mathcal{C}_j^t|}} \frac{2n}{|\mathcal{C}_j^t|} \varepsilon_1 + \frac{2\sqrt{d}\|V\|_{1,1}}{|\mathcal{C}_j^t|} \varepsilon_2.$$

It suffices to query the unitaries $U_I$ and $\mathcal{B}_I$ a number of $p = \widetilde{O}\left(\frac{\|V\|}{\sqrt{n}} \frac{k^{3/2}}{\varepsilon}\right)$ times within quantum multivariate mean estimator to get $\varepsilon_1 = O\left(\frac{\sqrt{n}}{\|V\|} \frac{\varepsilon}{k^{3/2}}\right)$. By the same toke, it suffices to query the unitaries $U_V$ and $\mathcal{B}_V$ a number of $q = \widetilde{O}\left(\frac{\|V\|_{1,1}}{n} \frac{k\sqrt{d}}{\varepsilon}\right)$ times within quantum multivariate mean estimator to get $\varepsilon_2 = O\left(\frac{n}{\|V\|_{1,1}} \frac{\varepsilon}{k\sqrt{d}}\right)$. This yields $\|c_j^{*t+1} - c_j^{t+1}\| \leq \varepsilon$ as wanted.

We now show how to perform the unitaries $U_V, U_I, \mathcal{B}_V, \mathcal{B}_I$. The binary-encoding unitary $\mathcal{B}_V$ is 1 quantum query ($O(\log n)$ time), followed by 1 controlled-SWAP on $kd$ qubits ($O(kd \log(kd))$ time (Berry et al., 2015)), while $\mathcal{B}_I$ is simply 1 controlled-SWAP on $k$ qubits. On the other hand, the

probability-distribution-encoding unitaries $U_V, U_I$ can be performed via the initial state preparations $|\bar{0}\rangle \mapsto \sum_{(i,l)\in[n]\times[d]} \sqrt{\frac{|V_{li}|}{\|V\|_{1,1}}}|i,l\rangle$ and $|\bar{0}\rangle \mapsto \frac{1}{\sqrt{n}}\sum_{i\in[n]}|i\rangle$, respectively, followed by the mapping $|i\rangle|\bar{0}\rangle \mapsto |i\rangle|\ell_i^t\rangle$. In Lemma 18 we show how to implement the mapping $|i\rangle|\bar{0}\rangle \mapsto |i\rangle|\ell_i^t\rangle$, where $\ell_i^t \in \{j \in [k] : \|v_i - c_j^t\|^2 \leq \min_{j'\in[k]}\|v_i - c_{j'}^t\|^2 + \nu\}$, using $\widetilde{O}\big(\frac{\|V\|_F}{\sqrt{n}}\frac{\sqrt{k}\|V\|_{2,\infty}}{\nu}\big)$ quantum queries and $\widetilde{O}\big(\frac{\|V\|_F}{\sqrt{n}}\frac{\sqrt{k}\|V\|_{2,\infty}}{\nu}\log n\big)$ time. In summary,

1. $\mathcal{B}_V$ requires $O(1)$ quantum queries and $O(\log(nd) + kd\log(kd))$ time;[5]

2. $U_V$ requires $\widetilde{O}\big(\frac{\|V\|_F}{\sqrt{n}}\frac{\sqrt{k}\|V\|_{2,\infty}}{\nu}\big)$ quantum queries and $\widetilde{O}\big(\frac{\|V\|_F}{\sqrt{n}}\frac{\sqrt{k}\|V\|_{2,\infty}}{\nu}\log n\big)$ time;

3. $\mathcal{B}_I$ requires no quantum queries and $O(k\log k)$ time;

4. $U_I$ requires $\widetilde{O}\big(\frac{\|V\|_F}{\sqrt{n}}\frac{\sqrt{k}\|V\|_{2,\infty}}{\nu}\big)$ quantum queries and $\widetilde{O}\big(\frac{\|V\|_F}{\sqrt{n}}\frac{\sqrt{k}\|V\|_{2,\infty}}{\nu}\log n\big)$ time.

Collecting all the term, the total number of quantum queries is $\widetilde{O}\big((p+q)\frac{\|V\|_F}{\sqrt{n}}\frac{\sqrt{k}\|V\|_{2,\infty}}{\nu}\big)$, while the overall time complexity is

$$\widetilde{O}\bigg((p+q)\bigg(\frac{\|V\|_F}{\sqrt{n}}\frac{\sqrt{k}\|V\|_{2,\infty}}{\nu}\log n + kd\bigg)\bigg)$$
$$= \widetilde{O}\bigg(\bigg(\sqrt{k}\frac{\|V\|}{\sqrt{n}} + \sqrt{d}\frac{\|V\|_{1,1}}{n}\bigg)\bigg(\frac{\|V\|_F\|V\|_{2,\infty}}{\sqrt{n}}\frac{k^{\frac{3}{2}}}{\varepsilon\nu}\log n + \frac{k^2 d}{\varepsilon}\bigg)\bigg). \qquad \square$$

**Lemma 18** (Approximate quantum cluster assignment). *Assume quantum query access to matrix $V = [v_1,\ldots,v_n] \in \mathbb{R}^{d\times n}$. Let $\delta \in (0,1)$, $\nu > 0$, and $0 < \varepsilon \leq \frac{\|V\|_F}{\sqrt{n}}$. Assume quantum query access to centroid matrix $C^t = [c_1^t,\ldots,c_k^t] \in \mathbb{R}^{d\times k}$ such that $\|c_j^t - |\mathcal{C}_j^{t-1}|^{-1}\sum_{i\in\mathcal{C}_j^{t-1}}v_i\| \leq \varepsilon$ with $|\mathcal{C}_j^{t-1}| = \Omega(\frac{n}{k})$ for all $j \in [k]$. There is a quantum algorithm that performs the mapping $|i\rangle|\bar{0}\rangle \mapsto |i\rangle|L_i^t\rangle$ such that, upon measuring $|L_i^t\rangle$ on the computational basis, the outcome equals $\ell_i^t \in \{j \in [k] : \|v_i - c_j^t\|^2 \leq \min_{j'\in[k]}\|v_i - c_{j'}^t\|^2 + \nu\}$ with probability at least $1 - \delta$. It uses $\widetilde{O}\big(\frac{\|V\|_F}{\sqrt{n}}\frac{\sqrt{k}\|V\|_{2,\infty}}{\nu}\big)$ quantum queries and $\widetilde{O}\big(\frac{\|V\|_F}{\sqrt{n}}\frac{\sqrt{k}\|V\|_{2,\infty}}{\nu}\log n\big)$ time, where $\widetilde{O}(\cdot)$ hides polylog factors in $k$, $d$, $\frac{1}{\delta}$, $\frac{1}{\nu}$, $\frac{\|V\|_F}{\sqrt{n}}$.*

*Proof.* We first describe how to perform the map $|i,j\rangle|\bar{0}\rangle \mapsto |i,j\rangle|w_{ij}\rangle$, where $|w_{ij} - \|v_i - c_j^t\|^2| \leq \frac{\nu}{2}$ with high probability. Recall from Definition 3 that we can perform the maps

$$\mathcal{O}_V : |i\rangle|\bar{0}\rangle \mapsto \sum_{l\in[d]} \frac{(v_i)_l}{\|v_i\|}|i,l\rangle \qquad \text{and} \qquad \mathcal{O}_{C^t} : |j\rangle|\bar{0}\rangle \mapsto \sum_{l\in[d]} \frac{(c_j^t)_l}{\|c_j^t\|}|j,l\rangle$$

in $O(\log(nd))$ time. Start then with the quantum state $|i,j\rangle\frac{|0\rangle+|1\rangle}{\sqrt{2}}|\bar{0}\rangle$ and perform the above maps controlled on the third register $\frac{|0\rangle+|1\rangle}{\sqrt{2}}$, i.e., perform $\mathcal{O}_V$ if the third register is $|0\rangle$ and $\mathcal{O}_{C^t}$ if it is $|1\rangle$. The final state is

$$\frac{1}{\sqrt{2}}|i,j\rangle\sum_{l\in[d]}\bigg(\frac{(v_i)_l}{\|v_i\|}|0,l\rangle + \frac{(c_j^t)_l}{\|c_j^t\|}|1,l\rangle\bigg).$$

After applying a Hadamard gate onto the third register, the state becomes

$$|i,j\rangle\sum_{l\in[d]}\bigg(\frac{1}{2}\bigg(\frac{(v_i)_l}{\|v_i\|} + \frac{(c_j^t)_l}{\|c_j^t\|}\bigg)|0,l\rangle + \frac{1}{2}\bigg(\frac{(v_i)_l}{\|v_i\|} - \frac{(c_j^t)_l}{\|c_j^t\|}\bigg)|1,l\rangle\bigg)$$
$$= |i,j\rangle(\sqrt{p_{ij}}|0\rangle|\psi_{ij}\rangle + \sqrt{1-p_{ij}}|1\rangle|\phi_{ij}\rangle),$$

---

[5]If one has access to a QRAG, then $\mathcal{B}_V$ requires only $O(\log(nd))$ time and the final runtime of Algorithm 4 is $\widetilde{O}\big(\big(\sqrt{k}\frac{\|V\|}{\sqrt{n}} + \sqrt{d}\frac{\|V\|_{1,1}}{n}\big)\frac{\|V\|_F\|V\|_{2,\infty}}{\sqrt{n}}\frac{k^{3/2}}{\varepsilon\nu}\log n + kd\big)$, where the term $\widetilde{O}(kd)$ comes from Footnote 3.

where

$$p_{ij} = \frac{1}{4} \sum_{l \in [d]} \left( \frac{(v_i)_l}{\|v_i\|} + \frac{(c_j^t)_l}{\|c_j^t\|} \right)^2 = \frac{1}{2} + \frac{\langle v_i, c_j^t \rangle}{2\|v_i\|\|c_j^t\|}$$

is the probability of measuring the third register on state $|0\rangle$, and $|\psi_{ij}\rangle$ and $|\phi_{ij}\rangle$ are "garbage" normalised states. It is then possible to apply a standard quantum amplitude estimation subroutine (Brassard et al., 2002) to obtain a quantum state $|i,j\rangle|\Psi'_{ij}\rangle$ such that, upon measuring onto the computation basis, the outcome $\widetilde{p}_{ij}$ is such that $|\widetilde{p}_{ij} - p_{ij}| \leq \frac{\nu}{4\|v_i\|\|c_j^t\|} \implies |w_{ij} - \|v_i - c_j^t\|^2| \leq \frac{\nu}{2}$ with probability at $1 - \delta_2$ for some $\delta_2 \in (0,1)$, where $w_{ij} = \|v_i\|^2 + \|c_j^t\|^2 - \|v_i\|\|c_j^t\|(2\widetilde{p}_{ij} - 1)$. It is then straightforward to obtain a new state $|\Psi_{ij}\rangle$ from $|\Psi'_{ij}\rangle$ which returns $w_{ij}$ upon measurement with probability $1 - \delta_2$. For each $(i,j) \in [n] \times [k]$, mapping $|i,j\rangle|0\rangle \mapsto |i,j\rangle|w_{ij}\rangle$ requires $O\left(\frac{\|v_i\|\|c_j^t\|}{\nu} \log \frac{1}{\delta_2}\right)$ quantum queries and $O\left(\frac{\|v_i\|\|c_j^t\|}{\nu} \log \frac{1}{\delta_2} \log(nd)\right)$ time.

Fix $i \in [n]$. The mapping $|i,j\rangle|\bar{0}\rangle \mapsto |i,j\rangle|w_{ij}\rangle$ can be viewed as quantum access to the vector $(w_{ij})_{j \in [k]}$. We thus employ the (variable-time) quantum minimum finding subroutine (Fact 13) in order to obtain the map $|i\rangle|\bar{0}\rangle \mapsto |i\rangle|L_i^t\rangle$, where upon measuring $|L_i^t\rangle$ on the computation basis, the outcome equals $\ell_i^t = \arg\min_{j \in [k]} w_{ij} \in \{j \in [k] : \|v_i - c_j^t\|^2 \leq \min_{j' \in [k]} \|v_i - c_{j'}^t\|^2 + \nu\}$ with probability $1 - \delta_1$. According to Fact 13, the query complexity of $|i\rangle|\bar{0}\rangle \mapsto |i\rangle|L_i^t\rangle$ is

$$O\left( \frac{\|v_i\|}{\nu} \log \frac{1}{\delta_1} \log \frac{1}{\delta_2} \sqrt{\sum_{j \in [k]} \|c_j^t\|^2} \right) = O\left( \frac{\|V\|_F}{\sqrt{n}} \frac{\sqrt{k}\|V\|_{2,\infty}}{\nu} \log \frac{1}{\delta_1} \log \frac{1}{\delta_2} \right), \qquad (2)$$

where we used that $\sum_{j \in [k]} \|c_j^t\|^2 = \|C^t\|_F^2 = O\left(k \frac{\|V\|_F^2}{n}\right)$ as in Lemma 11 and $\|v_i\| \leq \|V\|_{2,\infty}$, while the time complexity is $O(\log(nd))$ times the query complexity.

In order to analyse the success probability (see (Chen & de Wolf, 2023, Appendix A) for a similar argument), on the other hand, first note that we implement the unitary $\widetilde{U} : |i,j\rangle|\bar{0}\rangle \mapsto |i,j\rangle(\sqrt{1 - \delta_2}|w_{ij}\rangle + \sqrt{\delta_2}|w_{ij}^\perp\rangle)$, where $|w_{ij}\rangle$ contains the approximation $|w_{ij} - \|v_i - c_j^t\|^2| \leq \frac{\nu}{2}$ and $|w_{ij}^\perp\rangle$ is a normalised quantum state orthogonal to $|w_{ij}\rangle$. Ideally, we would like to implement $U : |i,j\rangle|\bar{0}\rangle \mapsto |i,j\rangle|w_{ij}\rangle$. Also

$$\forall|i,j\rangle : \qquad \|(U - \widetilde{U})|i,j\rangle|\bar{0}\rangle\| = \sqrt{(1 - \sqrt{1 - \delta_2})^2 + \delta_2} = \sqrt{2 - 2\sqrt{1 - \delta_2}} \leq \sqrt{2\delta_2},$$

using that $\sqrt{1 - \delta_2} \geq 1 - \delta_2$. Since (variable-time) quantum minimum finding does not take into account the action of $U$ onto states of the form $|i,j\rangle|\bar{0}^\perp\rangle$ for $|\bar{0}^\perp\rangle$ orthogonal to $|\bar{0}\rangle$, we can, without of loss of generality, assume that $\|U - \widetilde{U}\| \leq \sqrt{2\delta_2}$. The success probability of (variable-time) quantum minimum finding is $1 - \delta_1$ when employing the unitary $U$. However, since it employs $\widetilde{U}$ instead, the success probability decreases by at most the spectral norm of the difference between the "real" and the "ideal" total unitaries. To be more precise, the "ideal" (variable-time) quantum minimum finding is a sequence of gates $\mathcal{A} = U_1 E_1 U_2 E_2 \cdots U_N E_N$, where $U_i \in \{U, U^\dagger\}$, $E_i$ is a circuit of elementary gates, and $N$ is the number of queries to $U$, which can be upper-bounded as $O(\sqrt{k} \log \frac{1}{\delta_1})$. The "real" implementation, on the other hand, is $\widetilde{\mathcal{A}} = \widetilde{U}_1 E_1 \widetilde{U}_2 E_2 \cdots \widetilde{U}_N E_N$, where $\widetilde{U}_i \in \{\widetilde{U}, \widetilde{U}^\dagger\}$. Then $\|\mathcal{A} - \widetilde{\mathcal{A}}\| \leq N\|U - \widetilde{U}\| \leq N\sqrt{2\delta_2}$. The failure probability is thus $\delta_1 + N\sqrt{2\delta_2}$. By taking $\delta_1 = O(\delta)$ and $\delta_2 = O(\frac{\delta^2}{N^2})$, the success probability is $1 - \delta$. The final complexities are obtained by replacing $\delta_1$ and $\delta_2$ into Eq. (2). $\qquad\square$

## C LOWER BOUNDS

In this section we prove query lower bounds to the matrix $V = [v_1, \ldots, v_n] \in \mathbb{R}^{d \times n}$ for finding new centroids $c_1, \ldots, c_k \in \mathbb{R}^d$ given $k$ clusters $\{\mathcal{C}_j\}_{j \in [k]}$ that form a partition of $[n]$. We note that the task considered here is easier than the one performed by $(\varepsilon, \nu)$-$k$-means, since the clusters $\{\mathcal{C}_j\}_{j \in [k]}$ are part of the input. Nonetheless, query lower bounds for such problem will prove to be tight in most parameters. The main idea is to reduce the problem of approximating $\frac{1}{|\mathcal{C}_j|} \sum_{i \in \mathcal{C}_j} v_i$ for all $j \in [k]$ from the problem of approximating the Hamming weight of some bit-string, whose query complexity is given in the following well-known fact.

**Fact 19** ((Nayak & Wu, 1999, Theorem 1.11)). *Let $x \in \{0,1\}^n$ be a bit-string with Hamming weight $|x| = \Theta(n)$ accessible through queries. Consider the problem of outputting $w \in [n]$ such that $||x| - w| \leq m$ for a given $0 \leq m \leq n/4$. Its randomised classical query complexity is $\Theta(\min\{(n/m)^2, n\})$, while its quantum query complexity is $\Theta(\min\{n/m, n\})$.*

Before presenting our query lower bounds for $(\varepsilon, \nu)$-$k$-means, we shall need the following fact.

**Lemma 20.** *Given $x \in \mathbb{R}^d$ such that $\|x\|_1 \leq \varepsilon$ for $\varepsilon \geq 0$, there is $S \subseteq [d]$ with $|S| \geq \lceil \frac{d}{2} \rceil$ such that $|x_i| \leq \frac{2\varepsilon}{d}$ for all $i \in S$.*

*Proof.* Arrange the entries of $x$ is descending order, i.e., $|x_{k_1}| \geq |x_{k_2}| \geq \cdots \geq |x_{k_d}|$. Let $S = \{k_{\lfloor \frac{d}{2} \rfloor + 1}, \ldots, k_d\}$. Then $\varepsilon \geq \sum_{j=1}^{\lfloor d/2 \rfloor} |x_{k_j}| \geq \frac{d}{2}|x_i| \ \forall i \in S$, which implies $|x_i| \leq \frac{2\varepsilon}{d} \ \forall i \in S$. $\qquad \square$

**Theorem 21.** *Let $n, k, d \in \mathbb{N}$ and $\varepsilon > 0$. With entry-wise query access to $V = [v_1, \ldots, v_n] \in \mathbb{R}^{d \times n}$ and $(\|v_i\|_r)_{i \in [n]}$ for any $r \in [1, \infty]$ and classical description of partition $\{\mathcal{C}_j\}_{j \in [k]}$ of $[n]$ with $|\mathcal{C}_j| = \Omega(\frac{n}{k})$, outputting centroids $c_1, \ldots, c_k \in \mathbb{R}^d$ such that $\left\| c_j - \frac{1}{|\mathcal{C}_j|} \sum_{i \in \mathcal{C}_j} v_i \right\| \leq \varepsilon$ for all $j \in [k]$ has randomised and quantum query complexity $\Omega\left(\min\left\{ \frac{\|V\|_F^2}{n} \frac{kd}{\varepsilon^2}, nd \right\}\right)$ and $\Omega\left(\min\left\{ \frac{\|V\|_F}{\sqrt{n}} \frac{kd}{\varepsilon}, nd \right\}\right)$, respectively.*

*Proof.* Let $\{\mathcal{C}_j\}_{j \in [k]}$ be such that $\mathcal{C}_j = \{(j-1)\frac{n}{k} + 1, (j-1)\frac{n}{k} + 2, \ldots, j\frac{n}{k}\}$ for all $j \in [k]$. Consider the initial centroids $c_1^0, \ldots, c_k^0 \in \mathbb{R}^d$ defining $\{\mathcal{C}_j\}_{j \in [k]}$ as $c_j^0 = (0, 0, \ldots, 0, \frac{j}{k})$, i.e., its last entry is $\frac{j}{k}$. Now let $\alpha \in \mathbb{R}_+$ be a positive number to be determined later and $W := \{w \in \{0,1\}^d : w_d = 0 \text{ and } |w| = \lfloor \frac{d-1}{2} \rfloor\}$. Note that $\|w\|_r = \lfloor \frac{d-1}{2} \rfloor^{\frac{1}{r}}$ for all $w \in W$ and $r \in [1, \infty]$. Let then $V = [v_1, \ldots, v_n] \in \mathbb{R}^{d \times n}$ be such that, for each $j \in [k]$, the vectors $\{v_i\}_{i \in \mathcal{C}_j}$ are $v_i = \alpha c_j^0 + \alpha w_i$ (here the multiplication by $\alpha$ is done entry-wise), where $w_i$ is randomly picked from $W$. To be more precise, we pick the first $\lfloor \frac{d-1}{2} \rfloor$ bits of $w_i$ completely randomly and the next $\lfloor \frac{d-1}{2} \rfloor$ bits as the complement of the previous ones (plus $w_{d-1} = 0$ if $d$ is even, while $w_d = 0$ by definition). This means that the vectors $\{v_i\}_{i \in \mathcal{C}_j}$ belong to the $(d-1)$-sphere of diameter $\Theta(\alpha\sqrt{d})$ centered at $\alpha c_j^0$ and on the hyperplane orthogonal to $c_j^0$. Moreover, by construction, $\|v_i\|_r = \alpha\left(\frac{j^r}{k^r} + \lfloor \frac{d-1}{2} \rfloor\right)^{\frac{1}{r}}$ is constant for all $i \in \mathcal{C}_j$, so access to $\|v_i\|_r$ does not give any meaningful information about $c_j$. Now, notice that

$$\|V\|_F^2 = \alpha^2 \sum_{j \in [k]} \sum_{i \in \mathcal{C}_j} \|c_j^0 + w_i\|^2 \leq \alpha^2 \sum_{j \in [k]} \sum_{i \in \mathcal{C}_j} \frac{d+1}{2} = n\alpha^2 \frac{d+1}{2} \implies \alpha \geq \frac{\|V\|_F}{\sqrt{n}} \frac{\sqrt{2}}{\sqrt{d+1}}.$$

Assume we have an algorithm that outputs $c_1, \ldots, c_k \in \mathbb{R}^d$ such that $\left\| c_j - \frac{1}{|\mathcal{C}_j|} \sum_{i \in \mathcal{C}_j} v_i \right\| \leq \varepsilon$ for all $j \in [k]$. This allows us to output $\widetilde{w}_j := \frac{|\mathcal{C}_j|}{\alpha}(c_j - \alpha c_j^0) = \frac{n}{\alpha k}(c_j - \alpha c_j^0)$. Consider the first $\lfloor \frac{d-1}{2} \rfloor$ bits of $\widetilde{w}_j$ and $\sum_{i \in \mathcal{C}_j} w_i$ only. Then

$$\sum_{\ell=1}^{\lfloor (d-1)/2 \rfloor} \left| \widetilde{w}_{j\ell} - \sum_{i \in \mathcal{C}_j} w_{i\ell} \right| \leq \sqrt{\left\lfloor \frac{d-1}{2} \right\rfloor} \left\| \widetilde{w}_j - \sum_{i \in \mathcal{C}_j} w_i \right\| \leq \frac{n\varepsilon}{\alpha k} \sqrt{\left\lfloor \frac{d-1}{2} \right\rfloor} \qquad \forall j \in [k].$$

According to Lemma 20, for each $j \in [k]$ there is $S_j \subseteq [\lfloor \frac{d-1}{2} \rfloor]$ with $|S_j| \geq \lfloor \frac{d-1}{4} \rfloor$ such that $|\widetilde{w}_{j\ell} - \sum_{i \in \mathcal{C}_j} w_{i\ell}| \leq \frac{4n\varepsilon}{\alpha k\sqrt{d-1}}$ for $\ell \in S_j$, i.e., the number $\widetilde{w}_{j\ell}$ approximates $\sum_{i \in \mathcal{C}_j} w_{i\ell}$ up to additive error $\frac{4n\varepsilon}{\alpha k\sqrt{d-1}}$ for all $\ell \in S_j$ and $j \in [k]$. This means that we can approximate the Hamming weight of $k\lfloor \frac{d-1}{4} \rfloor$ *independent* bit-strings on $|\mathcal{C}_j| = \frac{n}{k}$ bits up to additive error $\frac{4n\varepsilon}{\alpha k\sqrt{d-1}}$ (the first $\lfloor \frac{d-1}{2} \rfloor$ bits of $w_i$ are independent by construction). According to Fact 19, the randomized and quantum query lower bounds for approximating $k\lfloor \frac{d-1}{4} \rfloor$ independent Hamming weights on $\frac{n}{k}$ bits each to precision $\frac{4n\varepsilon}{\alpha k\sqrt{d-1}}$ are, respectively,

$$\Omega\left( kd \min\left\{ \frac{n^2}{k^2} \frac{\alpha^2 k^2 d}{n^2 \varepsilon^2}, \frac{n}{k} \right\} \right) = \Omega\left( \min\left\{ \frac{\|V\|_F^2}{n} \frac{kd}{\varepsilon^2}, nd \right\} \right),$$

$$\Omega\left( kd \min\left\{ \frac{n}{k} \frac{\alpha k\sqrt{d}}{n\varepsilon}, \frac{n}{k} \right\} \right) = \Omega\left( \min\left\{ \frac{\|V\|_F}{\sqrt{n}} \frac{kd}{\varepsilon}, nd \right\} \right). \qquad \square$$

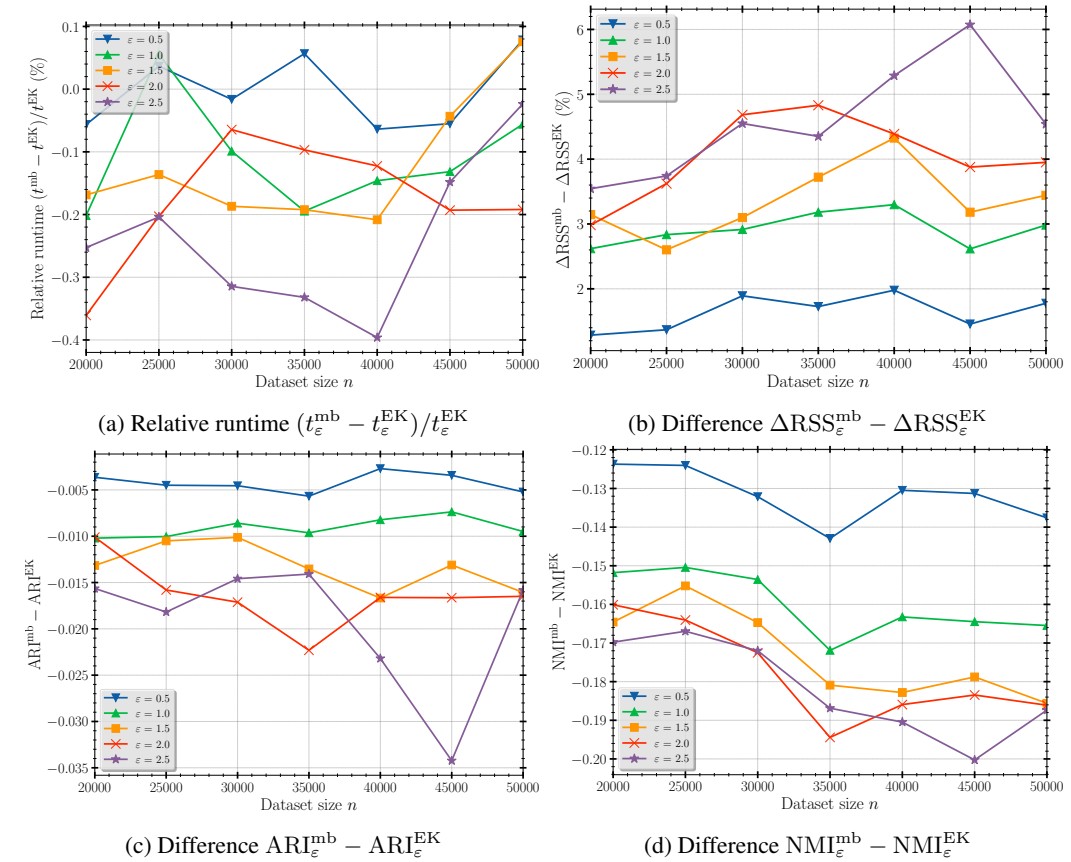

Figure 3: Comparison between `EKMeans` and mini-batch $k$-means algorithm in terms of runtime, relative residual sum of squares ($\Delta$RSS), adjusted Rand index (ARI), and normalized mutual information (NMI) as a function of the MNIST dataset size $n$. Here $k = 10$, $d = 784$, $\delta = 0.01$, and $\tau = 0.15$. Each point is the average of 85 random datasets and centroid initialisations.

# D  FURTHER NUMERICAL EXPERIMENTS

In this section, we conduct further numerical experiments that compare `EKMeans` and standard mini-batch algorithms, explore the dependence of `EKMeans` and standard $k$-means on the dimension $d$ and number of centroids $k$, and gauge the impact of sampling on the total runtime of `EKMeans`.

## D.1  COMPARING `EKMeans` AGAINST MINI-BATCH ALGORITHMS

As previously mentioned, `EKMeans` can be seen as a type of mini-batch $k$-means algorithm (Sculley, 2010; Bejarano et al., 2011), the main difference being that the quantities $\sum_{i \in \mathcal{C}_j^t} v_i$ are estimated by sampling vectors from the (pre-processed) distribution $\mathcal{D}^{(1)}_{(\|v_i\|)_{i=1}^n}$ instead of simply sampling $Q \subseteq [n]$ uniformly at random. As a consequence, `EKMeans` has a better sample and time complexity as proven in Theorems 8 and 9.

We now numerically validate such a result by comparing the performance of `EKMeans` and a mini-batch $k$-means algorithm wherein $Q \subseteq [n]$ is sampled uniformly at random. We employ the MNIST dataset composed of 60000 vectors in $[0,1]^d$ with $d = 784$. We pick $k = 10$ centroids, convergence threshold $\tau = 0.15$, approximation parameter $\varepsilon \in \{0.5, 1.0, 1.5, 2.0, 2.5\}$, and probability parameter $\delta = 0.01$. The sample sizes *for both algorithms* are $p = \left\lceil \frac{\|V\|^2}{n} \frac{k^2}{\varepsilon^2} \ln \frac{k}{\delta} \right\rceil$ and $q = \left\lceil \frac{\|V\|_{2,1}^2}{n^2} \frac{k^2}{\varepsilon^2} \ln \frac{k}{\delta} \right\rceil$. More precisely, we sample the *same amount* of points for `EKMeans` and the

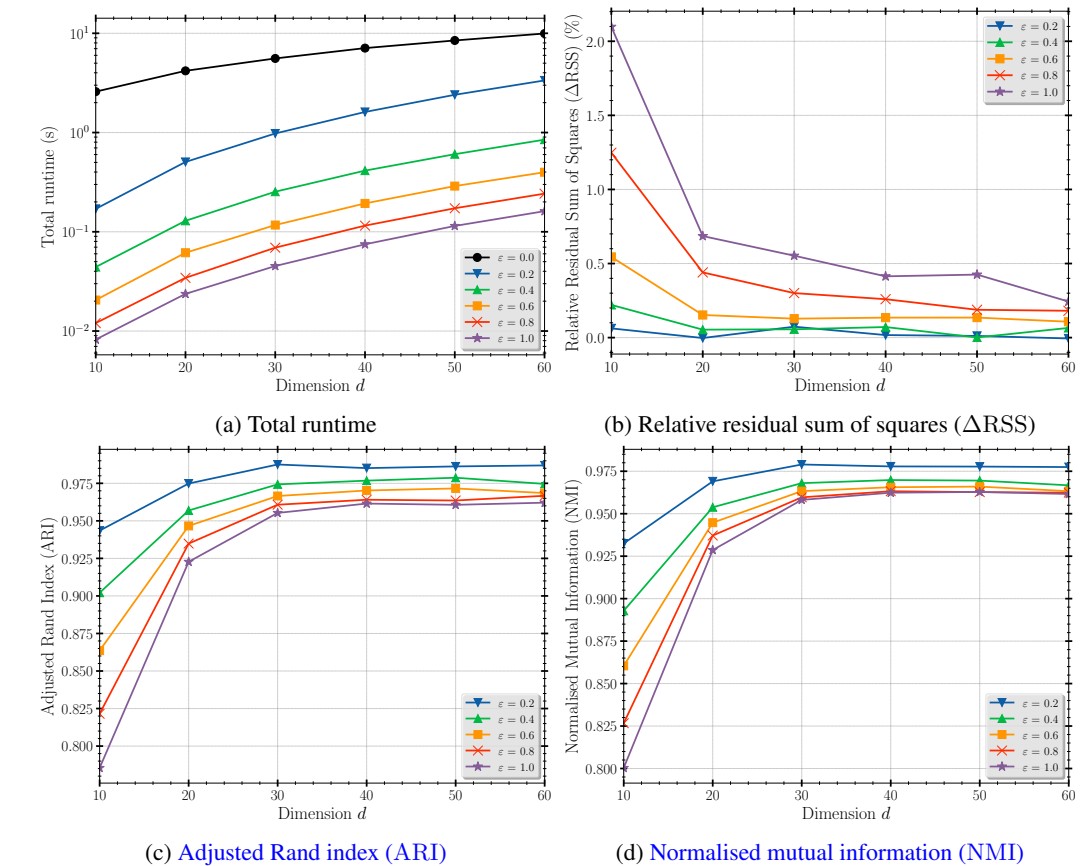

Figure 4: Total runtime, relative residual sum of squares ($\Delta$RSS), adjusted Rand index (ARI), and normalized mutual information (NMI) as a function of the dimension $d$ for synthetic dataset. Here $n = 300000$, $k = 5$, $\delta = 0.01$, and $\tau = 0.1$. The standard $k$-means is depicted as $\varepsilon = 0$. Each point is the average of 1750 random datasets and centroid initialisations.

mini-batch algorithm given a dataset $V$, which means that, according to Theorem 9, the mini-batch algorithm is "under-sampled".

Let $t_\varepsilon^{\mathrm{EK}}$ and $t_\varepsilon^{\mathrm{mb}}$ be the runtimes of EKMeans and the mini-batch algorithm with precision parameter $\varepsilon$, respectively. We compare the relative difference $(t_\varepsilon^{\mathrm{mb}} - t_\varepsilon^{\mathrm{EK}})/t_\varepsilon^{\mathrm{EK}}$ between runtimes in Figure 3a, which shows that, as expected, both algorithms have virtually the same time complexity in practice, with the mini-batch $k$-means being just slightly faster by less than $0.5\%$.

Now let $\Delta\mathrm{RSS}_\varepsilon^{\mathrm{EK}}$ and $\Delta\mathrm{RSS}_\varepsilon^{\mathrm{mb}}$ be the relative residual sum of squares of EKMeans and the mini-batch algorithm with precision parameter $\varepsilon$, respectively, and similarly for $\mathrm{ARI}_\varepsilon^{\mathrm{mb}}$, $\mathrm{ARI}_\varepsilon^{\mathrm{EK}}$, and $\mathrm{NMI}_\varepsilon^{\mathrm{mb}}$, $\mathrm{NMI}_\varepsilon^{\mathrm{EK}}$. In Figures 3b to 3d, we compare the performance of both algorithms via the differences $\Delta\mathrm{RSS}_\varepsilon^{\mathrm{mb}} - \Delta\mathrm{RSS}_\varepsilon^{\mathrm{EK}}$, $\mathrm{ARI}_\varepsilon^{\mathrm{mb}} - \mathrm{ARI}_\varepsilon^{\mathrm{EK}}$, and $\mathrm{NMI}_\varepsilon^{\mathrm{mb}} - \mathrm{NMI}_\varepsilon^{\mathrm{EK}}$. We can see that EKMeans is superior to the mini-batch algorithm in *all* metrics! The $\Delta$RSS of EKMeans is substantially *lower* by several percentage points (Figure 3b), meaning that its final centroids are better at minimising the residual sum of squares. On the other hand, EKMeans has a *higher* ARI (Figure 3c) and NMI (Figure 3d) compared to the mini-batch $k$-means (hence why the vertical axis is negative). The advantage in terms of the normalised mutual information is considerable, with a difference of $0.2$ in the worst case, while the adjusted Rand indexes are quite similar. The clustering of EKMeans is thus substantially closer to the standard $k$-means. All these results highlight the potential superiority of EKMeans over standard mini-batch $k$-means algorithms.

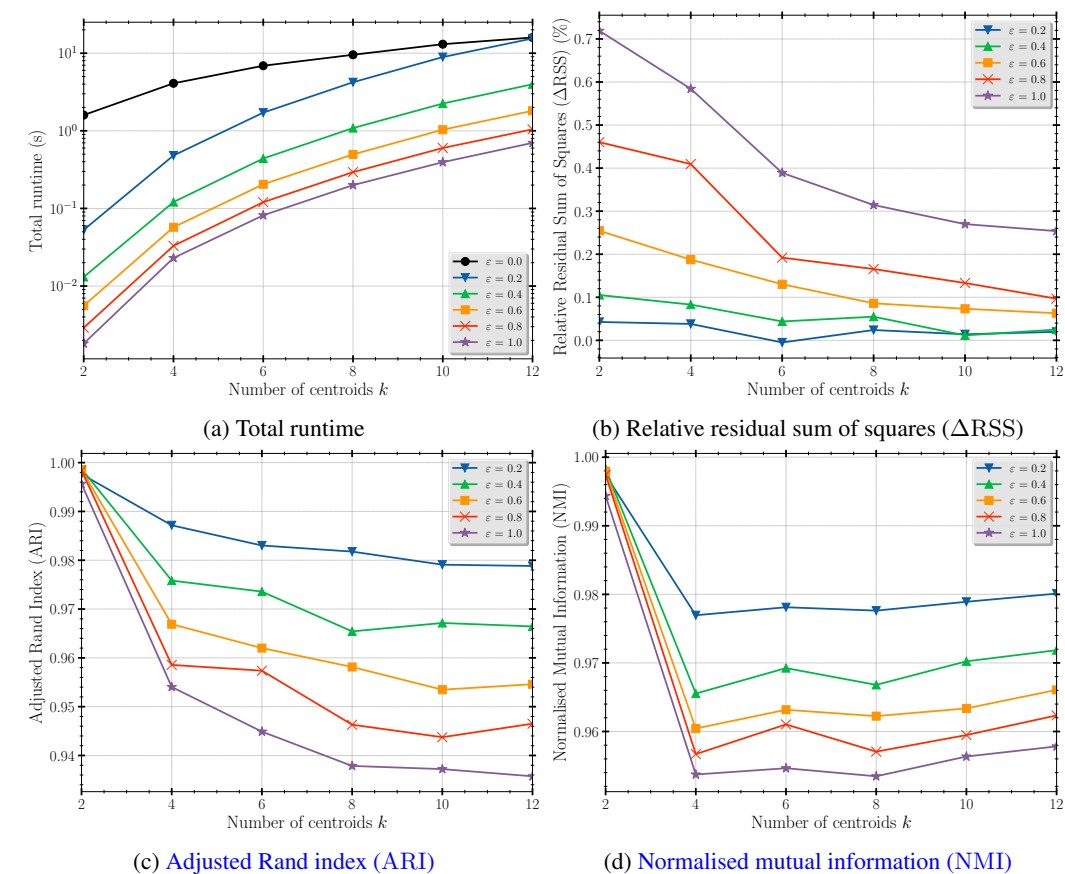

(a) Total runtime

(b) Relative residual sum of squares ($\Delta$RSS)

(c) Adjusted Rand index (ARI)

(d) Normalised mutual information (NMI)

Figure 5: Total runtime, relative residual sum of squares ($\Delta$RSS), adjusted Rand index (ARI), and normalized mutual information (NMI) as a function of the number of centroids $k$ for synthetic dataset. Here $n = 300000$, $d = 30$, $\delta = 0.01$, and $\tau = 0.1$. The standard $k$-means is depicted as $\varepsilon = 0$. Each point is the average of 400 random datasets and centroid initialisations.

## D.2 EXPLORING THE DEPENDENCE ON DIMENSION AND NUMBER OF CENTROIDS

In this section we conduct further numerical experiments exploring the dependence of EKMeans and standard $k$-means on the dimension $d$ (Figure 4) and number of centroids $k$ (Figure 5). All experiments were performed on the synthetic datasets from Section 4 as it is easier to change internal parameters like dimension $d$ and number of clusters $k$. We set the dataset size $n = 300000$, the convergence threshold $\tau = 0.1$, approximation parameter $\varepsilon \in \{0.2, 0.4, 0.6, 0.8, 1.0\}$, probability parameter $\delta = 0.01$. The samples sizes are once again $p = \left\lceil \frac{\|V\|^2}{n} \frac{k^2}{\varepsilon^2} \ln \frac{k}{\delta} \right\rceil$ and $q = \left\lceil \frac{\|V\|_{2,1}^2}{n^2} \frac{k^2}{\varepsilon^2} \ln \frac{k}{\delta} \right\rceil$.

Figure 4 collects our results regarding the total runtime, $\Delta$RSS, ARI, and NMI of EKMeans and $k$-means with respect to the dimension $d \in \{10, 20, 30, 40, 50, 60\}$. Here the number of centroids is fixed to $k = 5$. Once again, EKMeans is substantially faster than $k$-means for all dimensions as shown in Figure 4a, although the relative advantage is slightly smaller for larger $d$. As an example, for $d = 10$, $k$-means runs in $\approx 2.7$ s, while EKMeans with $\varepsilon = 1.0$ runs in $\approx 8$ ms, a $\approx 330$-fold improvement. For $d = 60$, this decreases to a $\approx 60$-fold advantage ($\approx 9.5$ s for $k$-means against $\approx 160$ ms for EKMeans with $\varepsilon = 1.0$). On the other hand, Figures 4b to 4d show, similarly to Figures 2b to 2d, that EKMeans returns good centroids compared to $k$-means across all dimensions as measured by $\Delta$RSS, ARI, and NMI.

In Figure 5 we explore the same properties — total runtime, RSS, and number of iterations — but now with respect to the number of centroids $k \in \{2, 4, 6, 8, 10, 12\}$. Here the dimension is fixed to $d = 30$. Figure 5a shows that EKMeans is still faster than $k$-means for all number of centroids. However, the advantage decreases as $k$ increases, a direct result of increasing the number of samples

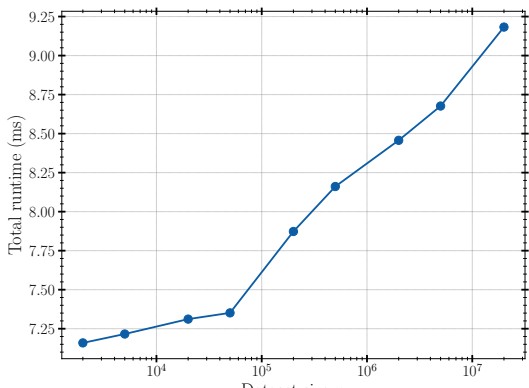

Figure 6: Total runtime as a function of the synthetic dataset size $n$. Here $k = 5$, $d = 5$, $\varepsilon = 0.5$, $\delta = 0.01$, and $\tau = 0.1$. Each point is the average of 255000 random datasets and centroid initialisations.

$p$ and $q$ quadratically with $k$. While the quadratic dependence of the number of samples on $k$ comes from rigorous theoretical results, $p$ and $q$ should obviously be capped at $n$ or even at a constant factor of $n$. The sample complexity of EKMeans can thus be made independent of $k$ for large values $k$ and its runtime follow the linear dependence from the standard $k$-means. On the other hand, Figures 5b to 5d show that once again EKMeans returns centroids with quality compared to $k$-means as measured by $\Delta$RSS, ARI, and NMI.

### D.3 THE IMPACT OF SAMPLING ON THE RUNTIME

As mentioned in Section 3, the runtime dependence on $n$ comes, at least theoretically, from sampling the sets of indices $P, Q \subseteq [n]$, being $O((p+q)\log n)$ under Definition 2. Such a dependence, if any, is hardly observed in Figure 2 given the different runtime scales. In Figure 6 we explore the runtime dependence on $n$ of EKMeans by covering a wide range of dataset sizes from $2 \cdot 10^3$ to $2 \cdot 10^7$. Once again we use the synthetic datasets from Section 4 and fix $k = 5$, $d = 5$, and $\varepsilon = 0.5$. The total runtime includes sampling the sets $P, Q$ and performing all iterations until convergence, i.e., until $\frac{1}{k}\sum_{j\in[k]}\|c_j^t - c_j^{t+1}\| \leq \tau = 0.1$. As can be observed, there is some dependence on $n$ coming from the sampling step, although quite small: a $10^4$-fold increase in the dataset set only adds a few milliseconds to the total clustering time. We note that sampling should be mostly independent of the dimension $d$, while the iterative clustering part is not. As a result, for larger $d$ the effect of sampling is even less pronounced compared to the total runtime. Ultimately, though, the dependence on $n$ (at least classically) is mostly an issue regarding how fast computers can access data in a RAM-like fashion. Sampling $P, Q$, specially $Q$, in our numerical experiments was done by converting a vector of floats into a distribution using the `discrete_distribution` function from the C++ library `random`. A more thorough analysis of sampling numbers from discrete distributions in C++ or other computational languages is beyond the scope of this work.

