# OpenReview forum: "Do you know what k-means? Clustering with constant number of samples"
_ICLR.cc/2026/Conference — Submitted to ICLR 2026_

### Official Review · Reviewer_7Cjf · 2025-10-30

**Soundness:** 3
**Presentation:** 2
**Contribution:** 2
**Rating:** 2
**Confidence:** 3

**Summary:**

The paper introduces ε-k-means, a classical approximation of Lloyd’s k-means algorithm that achieves exponential improvements in runtime dependence on data size, matching the efficiency of the q-means quantum algorithm by Kerenidis et al. (NeurIPS 2019). It also proposes an enhanced q-means algorithm that achieves a quadratic speedup over the new ε-k-means in several parameters. The quantum version relies only on QRAM-based state preparation rather than full quantum linear algebra primitives.

**Strengths:**

The paper addresses an interesting and worthwhile topic. The reasoning and justification behind the proposed approach appear well-founded and convincing.

**Weaknesses:**

The paper is difficult to follow due to its dense mathematical notation. The experimental evaluation is limited, as the proposed algorithm is compared to the original version only on synthetic datasets with fixed numbers of features and clusters. These parameters should have been varied to thoroughly assess effectiveness. Additionally, many standard synthetic datasets designed for clustering evaluation are omitted. The evaluation relies solely on RSS and runtime, neglecting other important metrics such as silhouette score, cluster sizes correlation, or external measures like Adjusted Rand Index (ARI) and Normalized Adjusted Rand Index (NARI) when ground truth labels are available (or against the original k-means result). Since each metric captures different aspects of clustering quality, this narrow evaluation is insufficient. Moreover, experiments involving quantum implementations are entirely missing. Finally, the paper omits relevant related work, including Poggiali, A., Berti, A., Bernasconi, A., Del Corso, G. M., & Guidotti, R. (2024). Quantum clustering with k-means: A hybrid approach. Theoretical Computer Science, 992, 114466.

**Questions:**

Questions can be derived from the above weaknesses.

---

> ### Author Response · Authors · 2025-11-23
>
> We thank the reviewer for their comments. Below we tried to address all their comments.
>
>
> **Technical clarity**: We really strove to prove very precise and rigorous statements about our algorithms. Therefore, it is unavoidable that some level of mathematical rigor and density is present. However, this is left entirely to the appendices and the main text is mostly free of heavy maths, describing the algorithms mostly on a high level. Unfortunately we don’t see how the proofs can be any simpler, but *we are happy to expand the manuscript with detailed intuitions behind the proofs and the algorithms*.
>
>
> **Experimental evaluation (metrics and dataset)**: In the updated version of the manuscript we now report the results on the MNIST dataset as per [Kerenidis et al., q-means, Neurips19], upon which we improve our quantum runtime. We also report the Adjusted Rand Index and Normalized Mutual Information of our algorithms. Extra experiments might require more time to conduct than is available during this rebuttal. The reason for initially considering only synthetic datasets was purely out of convenience, since it is easier to manipulate internal parameters like dimension and number of clusters within our synthetic datasets.
> *We hope it is clear now that  the new experiments on MNIST do not change the conclusions of this work.*
>
> **Experimental evaluation (other experiments)**
> We included in the experimental evaluation the metrics suggested by the reviewer. There are no other parameters that we can vary in order to assess the effectiveness of our algorithm as changing the numbers of features and clusters was already considered in Appendix D, which we assume the reviewer has read.
>
>
> **Quantum experiments**: It is very simple to see why experiments involving quantum implementations are entirely missing.  First, we would like to recall that quantum computers big enough to run our algorithms do not exist yet. Second, a cursory read of the quantum and classical algorithms would show that the output of the quantum algorithm (and its sensitivity to errors in distance estimation) can be fully inherited from the classical algorithm, making any further experiment on this superfluous. The only remaining experimental analysis of a quantum algorithm consists in a resource estimation, which clearly escapes the goal of our work (which, we recall, are theoretical guarantees for retrieving approximate solutions to an NP-hard problem with a time that does not depend on the number of samples).
>
> **Misc**: We are happy to cite the work mentioned by the reviewer.
>
> All in all, as we fully addressed the concerns and misunderstandings of our work, we ask the reviewer to raise their score.

---

### Official Review · Reviewer_yoqP · 2025-10-31

**Soundness:** 2
**Presentation:** 2
**Contribution:** 2
**Rating:** 2
**Confidence:** 4

**Summary:**

I am not familiar with quantum algorithms, so this review will only cover the classical results presented in the paper.

This paper presents a quantum-inspired algorithm for k-means. The authors claim that the running time of their algorithm achieves an exponential improvement in the dependence in n (number of data points) compared to classical k-means. They complement this with lower bounds claiming optimality for most parameters. Finally they experimentally compare their algorithm to the classical k-means algorithm.

**Strengths:**

The paper’s main strength is the introduction of another quantum-inspired algorithm, which enriches this promising research area.

**Weaknesses:**

There are several critical issues with this paper.

The running times as stated require a preprocessing step that requires $\tilde{O}(nd)$ time. So there is no “exponential speed up”.

The algorithm is essentially a mini-batch style algorithm, but there is no discussion of existing mini-batch approaches.

The experiments are lacking in several ways:
1) The paper only considers synthetic datasets - it should consider real world datasets such as mnist etc…
2) The paper only reports residual sum of squares - it should report ARI and NMI
3) The algorithm should be benchmarked against mini-batch k-means at the very least. Other approaches such as coresets would also be nice.
4) No code is provided

**Questions:**

See Weaknesses. Additional questions:
- The authors consider a version of k-means which converges if the movement of the centers falls below a certain threshold $\tau$. This is fine, but shouldn’t the parameter $\tau$ appear somewhere in the running time? Do your results work without this assumption?
- How does scaling the input vectors affect your running times?

---

> ### Author Response · Authors · 2025-11-24
>
> We thank the reviewer for their comments. We do find unfortunate, though, that the reviewer cannot assess the quantum components of our paper.
>
> **Speedups**: We obviously do not claim an exponential advantage if one includes the preprocessing step. Our results are clear in this direction: given a suitable sampling access to the data, Lloyd’s iteration can be approximated by sampling the input only a constant number of times. As the reviewer seems to be familiar with the literature in “quantum-inspired” algorithms, they will know that thanks to such results, quantum computer scientists cannot claim “exponential speedups” anymore. We believe that the community is not confused by the “preprocessing” + “algorithmic step”. The reviewer will agree that solely receiving the dataset (i.e., downloading from the internet, loading it in RAM) already constitutes a step that takes linear time in the size of the data. This is the time needed to perform our preprocessing.
>
> **Mini-batch**: We agree that our work can be seen as a kind of mini-batch k-means algorithm [Sculley-2010]. However, we are glad to have the opportunity to highlight the differences between our work and the previous literature in mini-batch algorithms. First, all previous works lack approximation guarantees for the centroids (which we have, along with lower bounds!), and cannot be used to perform sampling only once (during the preprocessing, as we do).
> Interestingly, we can follow the steps of our proof of Algorithm 1 to show approximation guarantees if the sampling is performed on a uniform distribution. It is possible to see that instead of $||V||_{2,1}^2$ we would obtain $n\sum |v_i|^2$, resulting in a much higher number of samples. Hence, as a mini-batch algorithm, our work shows much novelty:
> - First provable guarantees on centroid approximations
> - Shows a much better probability distribution for sampling
> - Lower bounds
> - The quantum version of the algorithm
> - No clusterability assumption
>
> We also have quantum+classical algos to mitigate the dependence on d, a result which is not part of the literature around k-means.
>
>
> **Stochastic algorithms**: The theoretical arguments on the convergence to the standard k-means [Convergence rate of stochastic k-means-ICAIS17], are hard to apply on batch k-means algorithm. In fact, the study of convergence of online k-means (batch size=1) has eluded the analysis for many years, with relatively recent results [Convergence of online k-means-ICAIS22] [Convergence rate of stochastic k-means-AISTAT17]. We believe that our work opens the door to much simpler analysis of convergence of stochastic k-means algorithms (see public answer of Poojan Chetan Shah). We will mention this as future work.
>
> **Clusterability** Furthermore we note that our work, along with the provable guarantees of k-means++ algorithm, has the chances to further relaxe the assumptions of [Clustering with spectral norm and the k-means algorithm-FOCS10] which is closing an open problem mentioned in [Spectral algorithms-TCS09]. In fact, our algorithm is not relying on any clusterability assumptions: we only assume that the size of the clusters is lower bounded by n/k, which can be removed by incurring a further factor in the runtime.
>
> **Experiments** In the updated version of the manuscript we now report the results on the MNIST dataset as per [q-means, Neurips19], upon which we improve our quantum runtime. We also report the ARI and NMI of our algorithms. The reason for initially considering only synthetic datasets was purely out of convenience: it is easier to manipulate internal parameters like dimension and number of clusters within our synthetic datasets. We believe that benchmarking our algorithms against other classical algorithms like mini-batch and coresets should be done in a follow-up work. *We hope it is clear that  the new experiments do not change the conclusions of this work.* You can find the code here: https://github.com/ForReviewers568/k-means
>
> **Threshold** we believe the reviewer is a bit confused about the role of $\tau$. The reported runtimes are for a single iteration of Lloyd’s algorithm. The total runtime is thus the runtime of a single iteration (i.e., the complexities reported in Table 1) times the number of iterations. It is obvious that the complexity of a single iteration should not depend on the total number of iterations and, consequently, on the threshold. Most classical iterative algorithms have the same hyperparameter (k-means, expectation maximization, mini-batch k-means...).
>
> **Scaling** If you scale the input, then the matrix norms will be scaled accordingly. On the other hand, the error parameter $\varepsilon$ will also be scaled accordingly. As a consequence, ratios like $\frac{\|V\|^2}{\varepsilon^2}$ will stay constant. Our complexities are thus invariant upon rescaling, similar to most algorithms.
>
> As we addressed the main concerns and misunderstandings, we ask the reviewer to raise their score.

---

> > ### Comment · Reviewer_yoqP · 2025-11-24
> >
> > Thank you for your detailed comments and for running additional experiments. Let me just address the most problematic point for now - from the experimental results provided it appears that if you were to compare to mini-batch k-means you will not achieve improved ARI \ NMI nor improved running times. I believe that any publishable version of your paper should definitely include these experiments.

---

> ### Author Response · Authors · 2025-11-28
>
> As suggested by the reviewer we included theoretical and numerical experiments on the performances of mini-batch k-means, as they are helping strengthen the reach and impact of our work. The numerical experiments can be found in Appendix D and show that $\mathtt{EKMeans}$ is superior to the mini-batch algorithm in **ALL** metrics, especially RSS and NMI. This confirms the relevance and impact of our theoretical results.
>
> Moreover, we proved a new result (Theorem 9) that shows that mini-batch k-means always requires more samples to obtain the same theoretical guarantees on the approximation of the centroids. To our knowledge, this is the first theoretical analysis of the guarantees of mini-batch k-means - only possible using the intuitions developed in our work.
>
>
> All in all, given that:
> - we clarified all the misunderstandings of the reviewer,
> - we addressed all the questions and weaknesses,
> - expanded our theoretical analysis to mini-batch algorithms as suggested,
> - extended the numerical results of such research area by proposing the first analysis of the approximation guarantees of minibatch k-means,
>
> Since we successfully addressed the most problematic point according to the reviewer, we kindly ask him or her to revise their score.
>
>
> In light of the reviewer’s stated difficulty in assessing the quantum algorithmic components of the paper, it may be appropriate to reflect this in the reported confidence score.

---

### Official Review · Reviewer_ge3j · 2025-11-01

**Soundness:** 2
**Presentation:** 2
**Contribution:** 3
**Rating:** 6
**Confidence:** 2

**Summary:**

This paper presents a classical $\varepsilon$-$k$-means (EKMeans)
algorithm that performs an exponential reduction in dependence on
data size $n$ while maintaining comparable clustering quality. The
algorithm estimates cluster sizes and centroids by drawing only a
constant number of sampled data points in each iteration, making the
time complexity of each iteration almost independent of $n$. In addition,
this paper presents an improved $q$-means quantum algorithm that
quadratically improves the runtime of EKMeans algorithm in several
parameters. This algorithm avoids complex quantum linear-algebra operations,
relying solely on QRAM access and quantum mean estimation to efficiently
update cluster centers. Numerical experiments show that EKMeans achieves
a significant speedup over the standard Lloyd's algorithm
on large-scale datasets while maintaining stable clustering quality.
This research is the first time to demonstrate. how the core ideas of quantum algorithms
can be transformed into efficient classical algorithms through dequantization.

**Strengths:**

1. This paper proposes a constant-sample, approximate $k$-means algorithm
(EKMeans) that maintains clustering quality while making the iteration
time almost independent of the data size $n$; and it also proposes
an improved quantum algorithm, forming a theoretical system that mutually
reinforces classical and quantum approaches.

2. $k$-means is one of the most commonly used clustering algorithms, so the related research has both theoretical and practical values.

3. The EKMeans algorithm in this paper provides a constant-time iterative
k-means version that can be implemented on conventional hardware without
relying on quantum hardware. It serves as a key example bridging QML
and classical randomized algorithms, deepening the understanding of
the boundaries between classical and quantum computing capabilities.

4. This paper is well structured and logically rigorous. It clearly defines
the problem and its background.

**Weaknesses:**

1. All experimental designs are based on synthetic data for comparison. Why not use real-world datasets?

2. The figures can be improved. E.g., the color scheme of the curves does not adequately distinguish certain parameter settings, such as  $\varepsilon=0.0$ vs. $\varepsilon=0.2$.

3. The title of the paper could be improved.

**Questions:**

1. In the experitmental section,  the authors use only synthetic data. What is the reason for not using real-world datasets?

2. Each iteration of k-means modifies the cluster partition (assigning different points to different clusters). Theoretically, each resampling iteration should reflect the latest cluster structure changes. However, if the same batch of P and Q is consistently used, these samples may primarily represent the structure at the initial partition. As the centroids gradually shift and boundaries adjust, the sampled points may no longer accurately represent the current cluster distribution, and the final updated centroids may be slow to react to the  real changes in some clusters. The algorithm may prematurely fall into a  locally stable solution (appearing convergent but with biases), especially when the initial sampling is not uniform or the cluster
distribution is complex. Do you have any theoretical guarantee or analysis against this issue?

---

> ### Author Response · Authors · 2025-11-23
>
> We thank the reviewer for their comments.
>
> **Question 1 (dataset choice)** :
> In the updated version of the manuscript we now report the results on the MNIST dataset as per [Kerenidis et al., q-means, Neurips19], upon which we improve our quantum runtime. We also report the Adjusted Rand Index and Normalized Mutual Information of our algorithms. The reason for initially considering only synthetic datasets was purely out of convenience, since it is easier to manipulate internal parameters like dimension and number of clusters within our synthetic datasets. *We hope it is clear now that  the new experiments on MNIST do not change the conclusions of this work.*
>
>
> **Question 2 (clarification on sampling)**:
> First, we want to mention that the sampling is independent of the cluster partitioning. Since the sampling is performed based on the norms of the vectors of the dataset, this is not affected at all by the cluster assignments of the centroids (see line 3 and 4 of Algorithm 1). There, the probability that the sampled points do not lead to a good enough approximation (compared to k-means) is bounded by $\delta$, and the sampling distribution does not depend on the cluster assignments.
>
> Using the union bound we can bound the failure probability that the initial sampling is not good enough for all $T$ iterations (assuming the number of iterations is bounded by T). For example: $T=50$ should be a reasonable bound for most of the dataset that are expected to be clustered with k-means. Concretely, assuming a failure probability of $\delta$, such an approach would only multiply the number of samples by $\log(T/\delta)/\log(1/\delta)$. This restores the theoretical guarantees of retrieving a good approximation of the centroids, while making the runtime of the iterations INDEPENDENT on $n$, up a cost of samples that is only logarithmic in the dataset size, and can be performed either when receiving the dataset (which has already a linear cost in the size) or during the preprocessing. **A** $10^4$ **-fold increase in the dataset set seize only adds a few milliseconds to the total clustering time, which remains constant**
>
> It is known that a bad selection of centroids affects k-means. For this reason k-means++ was created. If a good initialization of the centroids does not suffice, the solution is performing the algorithm again with a different selection of centroids. We do agree, however, that resampling the sets $P,Q$ might decrease the probability of the algorithm getting stuck in a local minima. We do not have any theoretical guarantees regarding this aspect since *we focused on improving Lloyd’s iterations, which is a heuristic algorithm for k-means.*
>
>
> We are happy to better clarify these observations in the manuscript, and we are also happy to change the title of the manuscript and make it more descriptive of the content. We will improve the quality of the figure in the next days.

---

> > ### Public Comment · ~Poojan_Chetan_Shah1 · 2025-11-24
> > **On getting stuck on local minima and related prior work.**
> >
> > I would like to point out to the reviewer that the work cited by the authors [1] study the k-means++ initialization problem in the quatum and quantum inspired setting. It is known that the k-means++ provides an O(log k) approximation guarantee to the k-means problem. Hence, running the QI-k-means++ algorithm of [1] followed by the accelerated lloyd's step proposed by the authors provide provable speedups having an O(log k) guarantee. Moreover, the runtime of [1] is still sublinear in the number of data points and depends upon the aspect ratio of the dataset (akin to the factor of ||V||_F^2/n which appears in this algorithm). Hence, initializing the proposed algorithm by QI-k-means++ doesnot undermine the runtime gains of the combined algorithm.
> >
> > I hope that this connection resolves the issue pointed out and that the above discussion adds to the discussion as well.
> >
> > [1] Poojan Chetan Shah and Ragesh Jaiswal. Quantum (inspired) D2
> > -sampling with applications. In
> > The Thirteenth International Conference on Learning Representations, 2025. URL https:
> > //openreview.net/forum?id=tDIL7UXmSS.

---

> > ### Comment · Reviewer_ge3j · 2025-11-26
> >
> > Thanks for the reply and for including new experimental results. I will keep my positive score.

---

### Author Response · Authors · 2025-12-01
**Summary and conclusions**

In light of recent events we would like to summarize our main contributions and the discussions with the reviewers. Our work proposes:

- A quantum and a classical algorithm (EQ-means and EK-means) for approximating the centroids of k-means, substantially improving the complexity of [Q-means - NeurIPS19] (and hence, by several orders of magnitude the runtime on real data)
- The **first analysis of the provable guarantees of classical mini-batch K-means** for centroid approximation.

- A quantum and a classical algorithm showing that it is possible to mitigate the dependence of the number of features $d$ in the runtime.

- Abundant numerical results (on real and synthetic data) showing the **superiority of EK-means against mini-batch algorithms (w.r.t all the metrics we studied)** and k-means (w.r.t the runtime).

- The **first analysis of quantum and classical lower bounds** for a Lloyd’s iteration (which shows that our algorithms are almost optimal).

- Thanks to our mathematical framework, thanks to a simple observation **we made the runtime of an iteration completely independent of $n$**. This trick requires a negligible overhead during the preprocessing of the data but retains our theoretical guarantees. In practice, our experiments show that we can cluster a dataset in time that does not depend on the number of samples, with only an additional cost of a few milliseconds that we spend when loading the data in memory.



During the conversation with the reviewers we discussed:
- how to improve the work with real datasets and not only synthetic datasets (which we did),
- that we can compare our work with mini-batch k-means (which we did, showing that our work is superior numerically and theoretically). This comparison was possible only thanks to our mathematical framework, which we extended to mini-batch k-means, giving the first theoretical approximation guarantees for this kind of algorithms,
- clarified a few misunderstandings of classical k-means and our work, and improved the quality of our presentation.



Our takeaway from this work is that it is possible to substantially improve the runtime of a  58-year old textbook algorithm while retaining similar approximation guarantees. Conversely, we showed that for the past 15 years mini-batch k-means algorithms were using a suboptimal probability distribution for sampling. We obtained improvements in classical algorithms by studying ideas born from the interplay of quantum and classical computation. Interestingly, to our knowledge this is the first practical application of the famous “dequantization” techniques mentioned by one of the reviewers, canceling yet-another “exponential speedup” of a quantum algorithm.

---

### Meta-Review · Area_Chair_MjXk · 2025-12-25

**Summary:**

This paper presents a quantum-inspired algorithm for the classic k-means problem. The paper is mostly a theoretical one, with bounds using only a constant number of samples. Examples were conducted on both synthetic data and MNIST data (after rebuttal).

Among the reviews, there are 2 negative scores (both score 2 - reject) and 1 weakly positive score. The following concerns were raised:
- In the initial submission, the experiments were only conducted on synthetic data.
- Only residual sum of squares was raised, and important metrics such as ARI and NMI weren't covered.
- The algorithm should be benchmarked against mini-batch k-means. Other approaches such as coresets would also be nice.

During the rebuttal, the authors addressed many points from the reviews. In particular, the authors conducted experiments on the MNIST data as well as behavior of parameters such as Adjusted Rand index (ARI) and Normalised mutual information (NMI). The contribution on the theory side is also clarified.

However, there are still perspectives that are not resolved:
- As pointed by Reviewer yoqP, it is not achieved improved ARI / NMI nor improved running times.
- The authors didn't address the point from Reviewer yoqP that benchmarks with other approaches such as coresets would also be nice.

In all, there exist doubts in the experimental results, and in theory there's another work by Chen et al. (2025) on the same topic. As a result, the paper is not ready enough for ICLR 2026.

**Reviewer Concerns:**

The authors replied and addressed many points. However, in general the results and experimental settings still have outstanding points -  see the metareview.

**Reviewer Scores:**

I don't think the scores will be significantly changed if the reviewers were able to participate fully in the discussion - reviewer ge3j kept positive attitude but didn't further increase the score, and yoqP also had follow-up questions about the experiments.

---

### Decision · Program_Chairs · 2026-01-26

Reject